# On the Role of Optimization in Double Descent: A Least Squares Study

**Ilja Kuzborskij**
DeepMind

**Csaba Szepesvári**
University of Alberta
DeepMind Edmonton, Canada

**Omar Rivasplata**
University College London

**Amal Rannen-Triki**
DeepMind

**Razvan Pascanu**
DeepMind

## Abstract

Empirically it has been observed that the performance of deep neural networks steadily improves with increased model size, contradicting the classical view on overfitting and generalization. Recently, the *double descent* phenomenon has been proposed to reconcile this observation with theory, suggesting that the test error has a second descent when the model becomes sufficiently overparameterized, as the model size itself acts as an implicit regularizer. In this paper we add to the growing body of work in this space, providing a careful study of learning dynamics as a function of model size for the least squares scenario. We show an excess risk bound for the gradient descent solution of the least squares objective. The bound depends on the smallest non-zero eigenvalue of the sample covariance matrix of the input features, via a functional form that has the double descent behaviour. This gives a new perspective on the double descent curves reported in the literature, as our analysis of the excess risk allows to decouple the effect of optimization and generalization error. In particular, we find that in the case of noiseless regression, double descent is explained solely by optimization-related quantities, which was missed in studies focusing on the Moore-Penrose pseudoinverse solution. We believe that our derivation provides an alternative view compared to existing works, shedding some light on a possible cause of this phenomenon, at least in the considered least squares setting. We empirically explore if our predictions hold for neural networks, in particular whether the spectrum of the sample covariance of features at intermediary hidden layers has a similar behaviour as the one predicted by our derivations in the least squares setting.

## 1 Introduction

Deep Neural Networks optimized by Gradient Descent (GD) methods have shown amazing versatility across a large range of domains. One of their most intriguing features is their ability to perform better with scale. Indeed, some of the most impressive results [see e.g. Brock et al., 2021, Brown et al., 2020, Senior et al., 2020, Schrittwieser et al., 2020, Silver et al., 2017, He et al., 2016 and references therein] have been obtained often by exploiting this fact, leading to neural network models that have at least as many parameters as the number of examples in the dataset they are trained on. Empirically, the limitation on the model size seems to be mostly imposed by hardware or compute. From a theoretical point of view, however, this property is quite surprising and counter-intuitive, as one would expect that in such extremely overparameterized regimes the learning would be prone to overfitting [Hastie et al., 2009, Shalev-Shwartz and Ben-David, 2014].

35th Conference on Neural Information Processing Systems (NeurIPS 2021).

Recently Belkin et al. [2019] proposed Double Descent (DD) phenomenon as an explanation. They argue that the classical view of overfitting does not apply in extremely overparameterized regimes, which were less studied prior to the emergence of the deep learning era. The classical view in the parametric learning models was based on error curves showing that the training error decreases monotonically when plotted against model size, while the corresponding test errors displayed a U-shape curve, where the model size for the bottom of the U-shape was taken to be a "sweet spot" [cf. Belkin et al., 2019, Fig. 1A] achieving an ideal trade-off between model size and generalization. Here, larger model sizes than the aforementioned ideal were thought to constitute the 'overfitting' regime, since the gap between the test error and the training error increased.

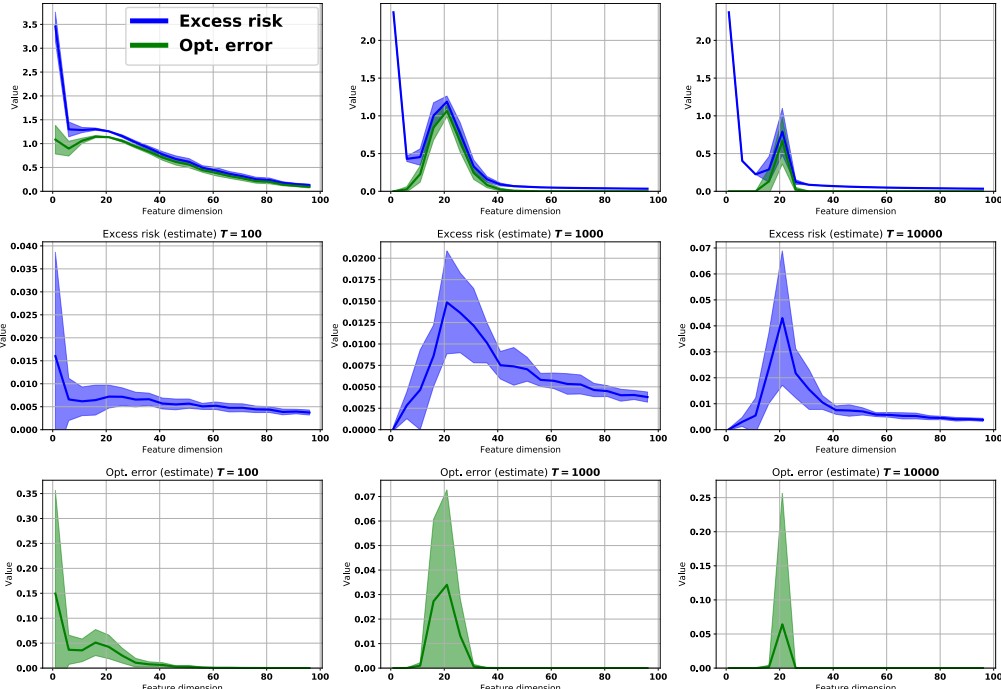

Figure 1: Evaluation of a synthetic setting inspired by Belkin et al. [2020]. We consider a linear regression problem ($n = 20$, $d \in [100]$), where regression parameters are fixed, and instances are sampled from a truncated (to $[-1, 1]$) normal density. GD is run with learning rate $\alpha = 0.05$ and initialization drawn from $\mathcal{N}(\mathbf{0}, \frac{1}{d}\mathbf{I})$. The first row demonstrates behavior of the excess risk (see Eq. (1)), the second shows an estimate of the excess risk (on $10^4$ held-out points), and the third shows an estimate of the optimization error.

The classical U-shape test error curve dwells in what is now called the under-parameterized regime, where the model size is smaller than the size of the dataset. Arguably, the restricted model sizes used in the past were tied to the available computing power. By contrast, it is common nowadays for model sizes to be larger than the amount of available data, which is called the over-parameterized regime. The divide between these two regimes is marked by a point where model size matches dataset size, which can be called the *interpolation threshold* [cf. Belkin et al., 2019, Fig. 1B].

The work of Belkin et al. [2019] argues that as model size grows beyond the interpolation threshold, one will observe a second descent of the test error that asymptotes in the limit to smaller values than those in the underparameterized regime, which indicates better generalization rather than overfitting. To some extent this was already known in the *nonparametric* learning where model complexity scales with the amount of data by design (such as in nearest neighbor rules and kernels), yet one can generalize well and even achieve statistical consistency [Györfi et al., 2002]. This has lead to a growing body of works trying to identify the mechanisms behind DD, to which the current manuscript belongs too. We refer the reader to Section 2, where the related literature is discussed. Similar to these works, our goal is also to understand the cause of DD. In this paper we explore a finite-time GD solution to least squares problems that allows us to work with analytic expressions for all the quantities involved. Fig. 1 provides a summary of our findings. In particular, it shows the behaviour of the *risk* (or the expected loss of the algorithm) in a linear regression setting with random inputs

and noise-free labels, for which in Section 3 we prove a bound that has the form

$$\left((1 - \alpha\widehat{\lambda}_{\min}^+)^{2T} + \frac{1}{\sqrt{n}}\right)\|\boldsymbol{w}^\star\|^2 \, .$$

Here labels are generated as $Y = \boldsymbol{X}^\top \boldsymbol{w}^\star$ for some unknown ground truth model $\boldsymbol{w}^\star \in \mathbb{R}^d$ and random inputs $\boldsymbol{X} \in \mathbb{R}^d$ on a unit sphere ($\|\boldsymbol{X}\|_{\boldsymbol{M}} = \sqrt{\boldsymbol{X}^\top \boldsymbol{M} \boldsymbol{X}} = 1$ for a positive matrix $\boldsymbol{M}$). The factor $\alpha$ is a constant learning rate, and $n$ is the number of examples in the training set. Note that the feature dimension $d$ coincides with the number of parameters in this particular setting, hence $d > n$ is the overparameterized regime. The quantity $\widehat{\lambda}_{\min}^+$ is *the smallest positive eigenvalue of the sample covariance matrix of the inputs*, and it is of special importance: We observe that the shape of the bound is controlled by $\widehat{\lambda}_{\min}^+$ via the exponential term, which increases as $\widehat{\lambda}_{\min}^+$ decreases while it decreases as $\widehat{\lambda}_{\min}^+$ grows, resulting in a peaked shape of the bound. In Fig. 1 we observe *a peaking behavior* not only in the risk but also in the quantity that we label 'optimization error' which is this special term of the risk bound that is purely related to optimization. The peaking behaviour of the risk—the Mean Squared Error (MSE) in case of the squared loss—was observed and studied in a number of settings [Belkin et al., 2019, Mei and Montanari, 2019, Derezinski et al., 2020] sometimes relating it to $\widehat{\lambda}_{\min}^+$, however, to the best of our knowledge the connection between the peaking behavior and optimization so far received less attention. In the absence of label noise, we conclude that DD manifests due to the optimization process. On the other hand, when label noise is present, in addition to the optimization effect, $\widehat{\lambda}_{\min}^+$ also has an effect on the generalization error.

**Our contributions:**  Our main theoretical contribution is presented in Section 3. In particular, Section 3.1 focuses on the noise-free least squares problem, Section 3.2 adds noise to the problem, and Section 3.3 deals with concentration of the sample-dependent $\widehat{\lambda}_{\min}^+$ around its population counterpart. The essential idea of the proof of our main result is outlined in Section 5. Sections 4 and 6 provide some discussion on the implications of our findings and an empirical exploration of the question whether simple neural networks have a similar behaviour than that of the least squares setting.

**Notation:**  The linear algebra and analysis notations used in this work are defined in Appendix A. We briefly mention here that we denote column vectors and matrices with small and capital bold letters, respectively, e.g. $\boldsymbol{\alpha} = [\alpha_1, \alpha_2, \ldots, \alpha_d]^\top \in \mathbb{R}^d$ and $\boldsymbol{A} \in \mathbb{R}^{d_1 \times d_2}$. Singular values of a rectangular matrix $\boldsymbol{A} \in \mathbb{R}^{n \times d}$ are denoted by $s_{\max}(\boldsymbol{A}) = s_1(\boldsymbol{A}) \geq \ldots \geq s_{n \wedge d}(\boldsymbol{A}) = s_{\min}(\boldsymbol{A})$. The rank of $\boldsymbol{A}$ is $r = \max\{k \mid s_k(\boldsymbol{A}) > 0\}$. The eigenvalues of a Positive Semi-Definite (PSD) matrix $\boldsymbol{M} \in \mathbb{R}^{d \times d}$ are non-negative and are denoted $\lambda_{\max}(\boldsymbol{M}) = \lambda_1(\boldsymbol{M}) \geq \ldots \geq \lambda_d(\boldsymbol{M}) = \lambda_{\min}(\boldsymbol{M})$, while $\lambda_{\min}^+(\boldsymbol{M})$ denotes the smallest *positive (non-zero)* eigenvalue.

## 2  Related Work

To the best of our knowledge DD was first mentioned by Vallet et al. [1989], who observed the phenomenon for the Moore-Penrose pseudo-inverse solutions (see [Loog et al., 2020] for a brief history). The recent literature revisited DD mainly considering ordinary least squares with the explicit solution given by the Moore-Penrose pseudo-inverse. These works have focused on instance-specific settings (making distributional assumptions on the inputs) while arguing when the analytic pseudo-inverse solutions yield DD behaviour [Belkin et al., 2020]. This was later extended to a more general setting showcasing the control of DD by the *spectrum* of the feature matrix [Derezinski et al., 2020]. In this paper we also argue that the spectrum of the covariance matrix has a critical role in DD, however we take into account the effect of GD *optimization*, which was missed by virtually all the previous literature due to their focusing on analytic solutions. The effect of the smallest non-zero eigenvalue on DD, through a condition number, was briefly noticed by Rangamani et al. [2020]. In this work we carry out a more comprehensive analysis and show how the excess risk of GD is controlled the smallest eigenvalue. In particular, $\widehat{\lambda}_{\min}^+$ has a U-shaped behaviour as the number of features increases, and we give a high-probability characterization of this behavior when inputs are subgaussian. To some extent, this is a non-asymptotic manifestation of the Bai-Yin law, whose connection to DD in an asymptotic setting was noted by Hastie et al. [2019, Theorem 1].

Some interest was also dedicated to the effect of bias and variance of DD [Mei and Montanari, 2019] in the same pseudo-inverse setting, while a more involved fine-grained analysis was later carried out

by Adlam and Pennington [2020]. In this work we focus on the influence of the optimization error, which is complementary to the bias-variance effects (typically we care about it once optimization error is negligible).

The DD behaviour was also observed beyond least squares, for instance in neural networks and other interpolating models [Belkin et al., 2019]. To some extent a formal connection to neural networks was first made by Mei and Montanari [2019] who studied the asymptotic behaviour of the risk under the random feature model, when $n, d^{\text{input}}, d^{\text{RF}} \to \infty$ while having $\frac{n}{d^{\text{input}}}$ and $\frac{d^{\text{RF}}}{d^{\text{input}}}$ fixed. Later on, with the popularity of the Neural Tangent Kernel (NTK) [Jacot et al., 2018] this connection became clearer as, within the NTK theory interpretation, shallow neural networks can be paralleled with kernelized predictors [Bartlett et al., 2021]. A detailed experimental study of DD in deep neural networks was carried out by [Nakkiran et al., 2019], who showed that various forms of regularization mitigate DD. Our results imply that $\ell_2$-regularization in least squares (ridge regression) makes $\widehat{\lambda}_{\min}^+$ well-controlled and therefore mitigates the peak. The same should carry over to the early-stopping in least squares [Yao et al., 2007] and to early stopping in shallow neural networks [Ji et al., 2021, Richards and Kuzborskij, 2021, Kuzborskij and Szepesvári, 2021].

In this work, we explain DD in least squares solution obtained by GD through the spectrum of the features, where optimization error has a visible role. While we do not present formal results for neural networks, we nevertheless empirically investigate on shallow neural networks whether our conclusions extend to neural nets as would be suggested by the NTK theory.

# 3    Excess Risk of the Gradient Descent Solution

We focus on the Gradient Descent (GD) algorithm, treated as a measurable map $\mathcal{A} : \mathcal{S} \times \mathbb{R}^d \to \mathbb{R}^d$ where $\mathcal{S} = \mathcal{Z}^n$ is the space of size-$n$ training sets. Consider a learning problem where hypotheses are parameterized by weight vectors $\boldsymbol{w} \in \mathbb{R}^d$. The risk of a given $\boldsymbol{w}$ is $L(\boldsymbol{w})$, and the empirical risk of $\boldsymbol{w}$ over a sample $S$ is $\widehat{L}_S(\boldsymbol{w})$. GD is applied to minimize the empirical risk. Given a training set $S \in \mathcal{S}$ and an initialization point $\boldsymbol{w}_0 \in \mathbb{R}^d$, we write $\mathcal{A}_S(\boldsymbol{w}_0)$ to indicate the output obtained recursively by the standard GD update rule with some fixed step size $\alpha > 0$, i.e. $\mathcal{A}_S(\boldsymbol{w}_0) = \boldsymbol{w}_T$, where

$$\boldsymbol{w}_t = \boldsymbol{w}_{t-1} - \alpha \nabla \widehat{L}_S(\boldsymbol{w}_{t-1}), \qquad t = 1, \ldots, T .$$

It is well-known that in the overparameterized regime ($d > n$), GD is able to achieve zero empirical loss. Therefore, rather than focusing on the generalization gap $L(\mathcal{A}_S(\boldsymbol{w}_0)) - \widehat{L}_S(\mathcal{A}_S(\boldsymbol{w}_0))$ it is more natural to compare the loss of $\mathcal{A}_S(\boldsymbol{w}_0)$ to that of the *best* possible predictor. Thus, we consider the *excess risk* defined as

$$\mathcal{E}_T = L(\mathcal{A}_S(\boldsymbol{w}_0)) - L(\boldsymbol{w}^\star) , \qquad \boldsymbol{w}^\star \in \arg\min_{\boldsymbol{w} \in \mathbb{R}^d} L(\boldsymbol{w}) . \tag{1}$$

## 3.1    Least Squares with Random Design and No Label Noise

We first consider a noise-free linear regression model with random inputs:

$$Y = \boldsymbol{X}^\top \boldsymbol{w}^\star$$

where the random input $\boldsymbol{X} \in \mathbb{R}^d$ is distributed according to some unknown distribution $P_X$ supported on a unit shpere ($\|\boldsymbol{X}\|_M = \sqrt{\boldsymbol{X}^\top \boldsymbol{M} \boldsymbol{X}} = 1$ for a positive definite matrix $\boldsymbol{M} \in \mathbb{R}^{d \times d}$). After observing a training sample $S = ((\boldsymbol{X}_i, Y_i))_{i=1}^n$, we run GD to minimize the empirical squared loss:

$$\widehat{L}_S(\boldsymbol{w}) = \frac{1}{2n} \sum_{i=1}^n (\boldsymbol{w}^\top \boldsymbol{X}_i - Y_i)^2 .$$

The sample covariance matrix is $\widehat{\boldsymbol{\Sigma}} = (\boldsymbol{X}_1 \boldsymbol{X}_1^\top + \cdots + \boldsymbol{X}_n \boldsymbol{X}_n^\top)/n$. Let $\widehat{\boldsymbol{\Sigma}} = \boldsymbol{U} \boldsymbol{S} \boldsymbol{V}^\top$ be the Singular Value Decomposition (SVD) with orthogonal matrices $\boldsymbol{U} = [\boldsymbol{u}_1, \ldots, \boldsymbol{u}_d]$ and $\boldsymbol{V} = [\boldsymbol{v}_1, \ldots, \boldsymbol{v}_d]$, and scaling matrix $\boldsymbol{S} = \mathrm{diag}(\widehat{\lambda}_1, \ldots, \widehat{\lambda}_d)$ where $\widehat{\lambda}_i = \lambda_i(\widehat{\boldsymbol{\Sigma}}) = s_i(\widehat{\boldsymbol{\Sigma}})$ are in the decreasing order: $\widehat{\lambda}_1 \geq \widehat{\lambda}_2 \geq \cdots \geq \widehat{\lambda}_d \geq 0$. The matrix $\widehat{\boldsymbol{\Sigma}}$ might be degenerate ($\widehat{\lambda}_d = 0$), and the non-degenerate part is given by $\boldsymbol{U}_r = [\boldsymbol{u}_1, \ldots, \boldsymbol{u}_r]$, $\boldsymbol{V}_r = [\boldsymbol{v}_1, \ldots, \boldsymbol{v}_r]$ and $\boldsymbol{S}_r = \mathrm{diag}(\widehat{\lambda}_1, \ldots, \widehat{\lambda}_r)$, where $r = \mathrm{rank}(\widehat{\boldsymbol{\Sigma}})$. We use the shorthand $\widehat{\lambda}_{\min}^+ = \lambda_{\min}^+(\widehat{\boldsymbol{\Sigma}}) = \lambda_r(\widehat{\boldsymbol{\Sigma}})$ for the minimal *positive* (*non-zero*) eigenvalue, and we denote $\widehat{\boldsymbol{M}} = \boldsymbol{U}_r \boldsymbol{U}_r^\top$. Note that $\widehat{\boldsymbol{M}}^2 = \widehat{\boldsymbol{M}}$. Now we state our main result in this setting.

**Theorem 1.** *For any $\boldsymbol{w}_0 \in \mathbb{R}^d$ and $x > 0$, with probability at least $1 - e^{-x}$ over inputs*

$$\mathcal{E}_T \leq (1 - \alpha \widehat{\lambda}_{\min}^+)^{2T} \|\boldsymbol{w}^\star - \boldsymbol{w}_0\|^2 + \frac{12}{\sqrt{n}} \left( \sqrt{\ln d} + \sqrt{x} \right) \left( \|\boldsymbol{w}_0\|^2 + 2\|\boldsymbol{w}^\star\|^2 \right) .$$

*Proof.* The theorem is a corollary of Theorem 2 below, whose proof is given in Appendix C. $\qquad\square$

Looking at Theorem 1, we can see that the excess risk is bounded by the sum of two terms. Note that the second term is negligible in many cases (consider the limit of infinite data) and additionally it is a term that remains constant during training as it does not depend on training data. Therefore, we are particularly interested in the first term of the bound, which is data-dependent and arises due to optimization error of the algorithm. This term increases as $\widehat{\lambda}_{\min}^+$ decreases and it decreases as $\widehat{\lambda}_{\min}^+$ grows, which results in a peaked shape. As we will see in the following Section 3.3, given mild assumptions on the input distribution, this is precisely the behaviour of $\widehat{\lambda}_{\min}^+$. Interestingly, only the first term in this bound is a source of the peak, which suggests that in a setting without the label noise, optimization error is the sole source of DD. Next we consider the case with label noise.

## 3.2 Least Squares with Random Design and Label Noise

Now, in addition to random inputs we introduce label noise into our model:

$$Y = \boldsymbol{X}^\top \boldsymbol{w}^\star + \varepsilon ,$$

where we have random variable $\varepsilon$, independent from $\boldsymbol{X}$, such that $\mathbb{E}[\varepsilon] = 0$ and $\mathbb{E}[\varepsilon^2] = \sigma^2$.

**Theorem 2.** *For any $\boldsymbol{w}_0 \in \mathbb{R}^d$ and $x > 0$, with probability at least $1 - e^{-x}$ over inputs*

$$\mathbb{E}[\mathcal{E}_T \mid \boldsymbol{X}_1, \dots, \boldsymbol{X}_n] \leq (1 - \alpha \widehat{\lambda}_{\min}^+)^{2T} \|\boldsymbol{w}^\star - \boldsymbol{w}_0\|^2 + \frac{4\sigma^2}{n} \left( \widehat{\lambda}_{\min}^+ \right)^{-2}$$
$$+ \frac{12}{\sqrt{n}} \left( \sqrt{\ln d} + \sqrt{x} \right) \left( \|\boldsymbol{w}_0\|^2 + 2\|\boldsymbol{w}^\star\|^2 + \sigma^2 \sum_{i=1}^{r} \widehat{\lambda}_i^{-1} \right) .$$

The proof is in Appendix C. Comparing to Theorem 1, the theorem gains two noise-dependent terms, namely $4(\sigma^2/n)(\widehat{\lambda}_{\min}^+)^{-2}$ and $\sigma^2(\widehat{\lambda}_1^{-1} + \cdots + \widehat{\lambda}_r^{-1})$. The former closely resembles the term controlling the generalization gap in the analysis of ridge regression [Shalev-Shwartz and Ben-David, 2014, Cor. 13.7], however, unlike ridge regression, here we have a dependence on the smallest non-zero eigenvalue instead of a regularization parameter. The latter is due to convergence of GD to the Moore-Penrose pseudo-inverse solution. This term can be further upper-bounded by $\sigma^2 r/\widehat{\lambda}_{\min}^+$, and so the last component in the bound is of order $(\|\boldsymbol{w}^\star\|^2 + \sigma^2 r/\widehat{\lambda}_{\min}^+)/\sqrt{n}$. Thus, as long as $\widehat{\lambda}_{\min}^+ \gtrsim r$, we should have a non-vacuous bound even in the overparameterized setting. In the next section we try to understand when this is the case by looking at the concentration of $\widehat{\lambda}_{\min}^+$.

## 3.3 Concentration of the Smallest Non-zero Eigenvalue

In this section we take a look at the behaviour of $\widehat{\lambda}_{\min}^+$ assuming that input instances $\boldsymbol{X}_1, \dots, \boldsymbol{X}_n$ are i.i.d. *random* vectors, sampled from some underlying marginal density that meets some regularity requirements (Definitions 1 and 2 below) so that we may use the results from random matrix theory [Vershynin, 2012]. Recall that $\widehat{\boldsymbol{\Sigma}} = (\boldsymbol{X}_1 \boldsymbol{X}_1^\top + \cdots + \boldsymbol{X}_n \boldsymbol{X}_n^\top)/n$ is the covariance matrix of the input features. We focus on the concentration of $\widehat{\lambda}_{\min}^+ = \lambda_{\min}^+(\widehat{\boldsymbol{\Sigma}})$ around its population counterpart $\lambda_{\min}^+ = \lambda_{\min}^+(\boldsymbol{\Sigma})$, where $\boldsymbol{\Sigma}$ is the population covariance matrix: $\boldsymbol{\Sigma} = \mathbb{E}[\boldsymbol{X}_1 \boldsymbol{X}_1^\top]$.

In particular, the Bai-Yin limit characterization of the extreme eigenvalues of sample covariance matrices [Bai and Yin, 1993] implies that $\widehat{\lambda}_{\min}^+$ has almost surely an asymptotic behavior $(1 - \sqrt{d/n})^2$ as the dimensions grow to infinity, assuming that the matrix $\boldsymbol{D}^\top := [\boldsymbol{X}_1, \dots, \boldsymbol{X}_n] \in \mathbb{R}^{d \times n}$ has independent entries. We are interested in the non-asymptotic version of this result. However, unlike Bai and Yin [1993], we do not assume independence of all entries, but rather independence of observation vectors (columns of $\boldsymbol{D}^\top$). This will be done by introducing a distributional assumption: we assume that observations are *sub-Gaussian* and *isotropic* random vectors.

**Definition 1** (Sub-Gaussian random vectors). *A random vector $\boldsymbol{X} \in \mathbb{R}^d$ is sub-Gaussian if the random variables $\boldsymbol{X}^\top \boldsymbol{y}$ are sub-Gaussian for all $\boldsymbol{y} \in \mathbb{R}^d$. The sub-Gaussian norm of a random vector $\boldsymbol{X} \in \mathbb{R}^d$ is defined as*

$$\|\boldsymbol{X}\|_{\psi_2} = \sup_{\|\boldsymbol{y}\|=1} \sup_{p \geq 1} \left\{ \frac{1}{\sqrt{p}} \mathbb{E}[|\boldsymbol{X}^\top \boldsymbol{y}|^p]^{\frac{1}{p}} \right\} .$$

**Definition 2** (Isotropic random vectors). *A random vector $\boldsymbol{X} \in \mathbb{R}^d$ is called isotropic if its covariance is the identity: $\mathbb{E}[\boldsymbol{X}\boldsymbol{X}^\top] = \boldsymbol{I}$. Equivalently, $\boldsymbol{X}$ is isotropic if $\mathbb{E}[(\boldsymbol{X}^\top \boldsymbol{x})^2] = \|\boldsymbol{x}\|^2$ for all $\boldsymbol{x} \in \mathbb{R}^d$.*

Let $\boldsymbol{\Sigma}^\dagger$ be the Moore-Penrose pseudoinverse of $\boldsymbol{\Sigma}$. In Appendix D we prove the following.[1]

**Lemma 1** (Smallest non-zero eigenvalue of sample covariance matrix). *Let $\boldsymbol{D}^\top = [\boldsymbol{X}_1, \ldots, \boldsymbol{X}_n] \in \mathbb{R}^{d \times n}$ be a matrix with i.i.d. columns, such that $\max_i \|\boldsymbol{X}_i\|_{\psi_2} \leq K$, and let $\widehat{\boldsymbol{\Sigma}} = \boldsymbol{D}^\top \boldsymbol{D}/n$, and $\boldsymbol{\Sigma} = \mathbb{E}[\boldsymbol{X}_1 \boldsymbol{X}_1^\top]$. Then, for every $x \geq 0$, with probability at least $1 - 2e^{-x}$, we have*

$$\lambda^+_{\min}(\widehat{\boldsymbol{\Sigma}}) \geq \lambda^+_{\min}(\boldsymbol{\Sigma}) \left( 1 - K^2 \left( c\sqrt{\frac{d}{n}} + \sqrt{\frac{x}{n}} \right) \right)^2_+ \qquad \text{for } n \geq d ,$$

*and furthermore, assuming that $\|\boldsymbol{X}_i\|_{\boldsymbol{\Sigma}^\dagger} = \sqrt{d}$ a.s. for all $i \in [n]$, we have*

$$\lambda^+_{\min}(\widehat{\boldsymbol{\Sigma}}) \geq \lambda^+_{\min}(\boldsymbol{\Sigma}) \left( \sqrt{\frac{d}{n}} - K^2 \left( c + 6\sqrt{\frac{x}{n}} \right) \right)^2_+ \qquad \text{for } n < d ,$$

*where we have an absolute constant $c = 2^{3.5}\sqrt{\ln(9)}$.*

Lemma 1 is a non-asymptotic result that allows us to understand the behaviour of $\widehat{\lambda}^+_{\min}$, and hence the behaviour of the excess risk bound that depends on this quantity, for fixed dimensions. We will exploit this fact in the following section in which we discuss the implications of our findings.

## 4 Discussion

The above concentration inequalities for $\widehat{\lambda}^+_{\min}$ combined with Theorem 1, in the *overparameterized* regime ($d > n$) give us [2]

$$\mathcal{E}_T \lesssim \left( \left( 1 - \frac{\alpha}{n}(\sqrt{d} - \sqrt{n} - 1)^2_+ \right)^{2T} + \frac{1}{\sqrt{n}} \right) \|\boldsymbol{w}^\star\|^2 .$$

A similar bound holds in the *underparameterized* case ($d < n$) but replacing the term $(\sqrt{d} - \sqrt{n} - 1)^2_+$ with $(\sqrt{n} - \sqrt{d} - 1)^2_+$. Note that the term multiplying the learning rate is $(\sqrt{d/n} - 1 - 1/\sqrt{n})^2$, in accordance with the Bai-Yin limit which says that asymptotically $\widehat{\lambda}^+_{\min} \sim (\sqrt{d/n} - 1)^2$. It is interesting to see how $\left( 1 - \alpha(\sqrt{d/n} - 1)^2_+ \right)^{2T}$ varies with model size $d$ for a given fixed dataset size $n$ and fixed number of gradient updates $T$. Setting $\gamma = d/n$ and considering the cases $\gamma \to 0$ (underparameterized regime), $\gamma \sim 1$ (the peak), and $\gamma > 1$ (overparameterized regime) it becomes evident that this term has a double descent behaviour. Thus, the double descent is captured in the part of the excess risk bound that corresponds to learning dynamics on the space spanned by $\widehat{\boldsymbol{M}}$.

Similarly, we can now consider the scenario with label noise: we can similarly bound the excess risk, following the same logic as for noise-free case; however we have an additional dependence on $\sigma^2$ via the term $4(\sigma^2/n)(\widehat{\lambda}^+_{\min})^{-2}$. While this does not interfere with the double descent shape as we change model size, it does imply that the peak is dependent on the amount of noise: The more noise in the learning problem, the larger we expect the peak at the interpolation threshold to be.

While the presence of the double descent has been studied by several works considering pseudoinverse solutions, our derivation provides potentially new interesting insights. Optimization plays a role in the

---

[1]$(x)_+ = \max \{0, x\}$

[2]We use $f \lesssim g$ when there exists a universal constant $C > 0$ such that $f \leq Cg$ uniformly over all arguments.

appearance of DD and its impact depends on the strength of the label noise. In the noise-free setting, DD curves seem to be solely due to optimization. In the presence of noise, there is a relationship between the strength of label noise and the shape of the DD curve: The larger the noise is, the larger is the peak in DD curve. This agrees with the typical intuition in the underparmetrized regime that the model fits the noise when it has enough capacity, leading towards a spike in test error. However, due to the dependence on $\widehat{\lambda}_{\min}^+$, it is subdued as the model size grows.

**Connection to neural networks.** Taking some inspiration from our observations for least squares, it is natural to ask whether we can observe a similar effect in neural networks trained by GD. One argument here could be to decompose learning in neural networks into a feature extraction and linear prediction components, and to check whether $\widehat{\lambda}_{\min}^+$ of features obtained from the intermediary layers could exhibit the behaviour leading to the double descent. This in particular might be significant if we think of the last layer of the architecture as a least squares problem (assuming we are working with mean squared error), and all previous layers as some random projection, ignoring that learning is affecting this projection as well. It is also interesting to investigate the behaviour of $\widehat{\lambda}_{\min}^+$ in combination with techniques used by practitioners, such as batch-norm [Ioffe and Szegedy, 2015], layer-norm [Ba et al., 2016], and residual connections [He et al., 2016], which are typically used in recent architectures, ensuring that features of intermediary layers are well "conditioned". In Section 6 we empirically look at some of these questions.

From a theoretical point of view, one possibility for adapting our results to neural network learning could be to consider a Neural Tangent Kernel (NTK) framework [Jacot et al., 2018]. Interestingly, this may require results on the concentration of the neural tangent random feature covariance matrix. For shallow networks the concentration phenomena for NTK matrices are expected to be similar as presented in our work [Tropp, 2012]. For deep networks, the spectrum of an NTK matrix is not well-understood and this is an active area of research itself [Nguyen et al., 2021] with open questions, which would be interesting to explore in the future work.

## 5   Proof idea

The proof of our main result (Theorem 2) rests upon the observation that under GD optimizing an overparameterized least squares objective, the iterates span a low-dimensional subspace given by the eigenvectors of a sample covariance matrix. This is captured by the following straightforward decomposition (see Appendix C for the rest of the proof):

**Proposition 1.** *For any $\boldsymbol{w}_0 \in \mathbb{R}^d$,*

$$\mathcal{E}_T \le \underbrace{\|\mathcal{A}_S(\boldsymbol{w}_0) - \mathcal{A}_S(\boldsymbol{w}^\star)\|_{\widehat{\boldsymbol{\Sigma}}}^2}_{(1)} + \underbrace{\|\mathcal{A}_S(\boldsymbol{w}^\star) - \boldsymbol{w}^\star\|_{\widehat{\boldsymbol{\Sigma}}}^2}_{(2)} + \underbrace{\|\boldsymbol{\Sigma} - \widehat{\boldsymbol{\Sigma}}\|_2 \left(\|\mathcal{A}_S(\boldsymbol{w}_0)\|^2 + \|\boldsymbol{w}^\star\|^2\right)}_{(3)} \,.$$

Here, the first term $(1)$ vanishes as long as the algorithm-map $\mathcal{A}_S$ is contractive on the aforementioned subspace, which we show in Appendix C.1 thanks to the closed-form expression of GD iterates.

Term $(2)$ captures algorithm's sensitivity to the label noise: How far would GD go when initialized at the *minimum of the risk*? Indeed, it is easy to see that when there is no label noise, term $(2)$ is zero (one can demonstrate it by the descent lemma). In the presence of noise, matters are more complicated, and we employ a more or less standard technique where we recursively track the distance between GD iterates and "virtual" iterates, which are obtained as if we could remove the noise. This results in a bound $4(\sigma^2/n)(\widehat{\lambda}_{\min}^+)^{-2}$, which is shown in Appendix C.3.

Finally, term $(3)$ is essentially a concentration of the sample covariance matrix and ensuring that the norm of the solution remains well-behaved, see Appendix C.2. The concentration of the sample covariance matrix is due to the matrix Chernoff inequality [Tropp, 2012]. We control the norm of the solution by relating it to the Moore-Penrose pseudoinverse solution which can be written in a closed-form expression. This makes it easy to see that when the label noise is absent, the norm depends only on $\boldsymbol{w}^\star$ and $\boldsymbol{w}_0$. On the other hand, when the noise is present the behaviour of the bound will depend on behaviour of $\widehat{\lambda}_{\min}^+$ as discussed in Section 3.2.

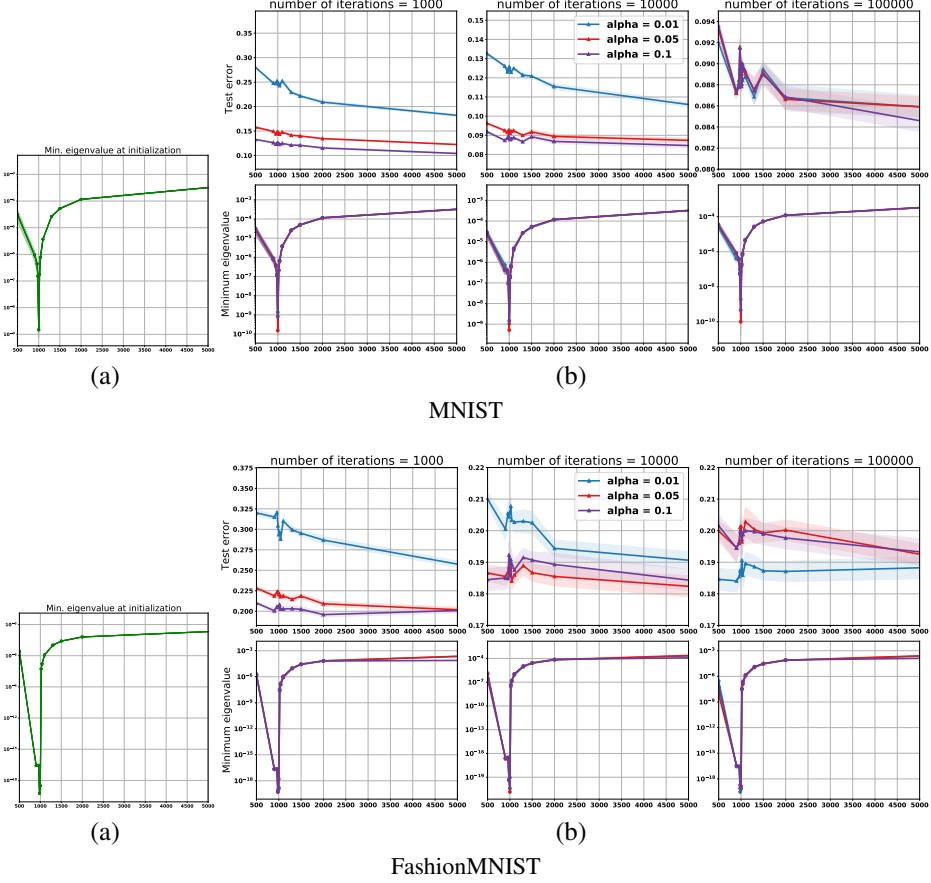

Figure 2: Training one hidden layer networks of increasing width on MNIST (top) and FashionMNIST (bottom): (a) Minimum positive eigenvalue of the intermediary features at initialization - (b) Test error and corresponding minimum eigenvalue of the intermediary features at different iterations

## 6  Empirical exploration in neural networks

The first natural question to ask is whether the observed behaviour for the least squares problem is reflected when working with neural networks. To explore this hypothesis, and to allow tractability of computing various quantities of interest (like $\widehat{\lambda}_{\min}^{+}$), we focus on one hidden layer MLPs on the MNIST and FashionMNIST datasets. We follow the protocol used by Belkin et al. [2019], relying on a squared error loss. In order to increase the model size we simply increase the dimensionality of the latent space, and rely on gradient descent with a fixed learning rate and a training set of 1000 randomly chosen examples for both datasets. More details can be found in Appendix G.

Fig. 2 provides the main findings on this experiment. Similar to the Fig. 1, we depict 3 columns showing snapshots at different number of gradient updates: 1000, 10000 and 100000. The first row shows test error (number of miss-classified examples out of the test examples) computed on the full test set of 10000 data points which as expected shows the double descent curve with a peak around 1000 hidden units. Note that the peak is relatively small, however the behaviour seems consistent under 5 random seeds for the MNIST experiment.[3] The second row and potentially the more interesting one looks at the $\widehat{\lambda}_{\min}^{+}$ computed on the covariance of the activations of the hidden layer, which as predicted by our theoretical derivation shows a dip around the interpolation threshold, giving the expected U-shape. Even more surprisingly this shape seems to be robust throughout learning, and the fact that the input weights and biases are being trained seems not to alter it, thus suggesting that our derivation might provide insights in the behaviour of deep models.

---

[3]The error bars for the test error in all the other experiments are estimated by splitting the test set into 10 subsets.

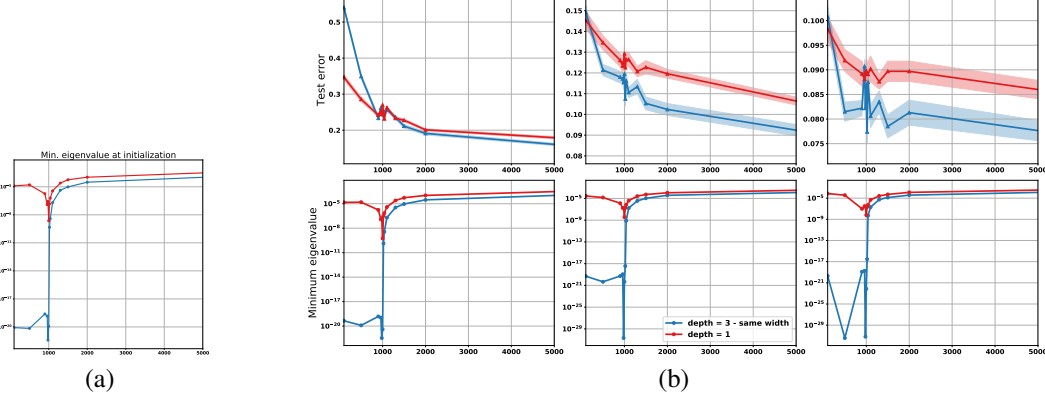

Figure 3: Training networks of increasing width with 1 and 3 hidden layers on MNIST: (a) Minimum positive eigenvalue of the intermediary features at initialization - (b) Test error and corresponding minimum eigenvalue of the intermediary features at different iterations

Following this, if we think of the output layer as solving a least squares problem, while the rest of the network provides a projection of the data, we can consider what can affect the conditioning of the last latent space of the network. We put forward the hypothesis that $\widehat{\lambda}_{\min}^+$ is not simply affected by the number of parameters, but actually the distribution of these parameters in the architecture matters.

To test this hypothesis, we conduct an experiment where we compare the behavior of a network with a single hidden layer and a network with three hidden layers. For both networks, we increase the size of the hidden layers. For the deeper network, we consider either increasing the size of all the hidden layers or grow only the last hidden layer while keeping the others to a fixed small size, creating a strong bottleneck in the network. Fig. 3 shows the results obtained with the former, while the effect of the bottleneck can be seen in Appendix F. We first observe that for the three tested networks, the drop in the minimum eigenvalues happens when the size of the last hidden layer reaches the number of training samples, as predicted by the theory. The magnitude of this drop and behavior across the different tested sizes depends however on the previous layers. In particular, we observe that the bottleneck yields features that are more ill-conditioned than the network with wide hidden layers, where the width of the last layer on its own can not compensate for the existence of the bottleneck. Moreover, from Fig. 3, we can clearly see that the features obtained by the deeper network have a bigger drop in the minimum eigenvalue, which results, as expected in a higher increase in the test error around the interpolation threshold.

It is well known that depth can harm optimization making the problem ill-conditioned, hence the reliance on skip-connections and batch normalization [De and Smith, 2020] to train very deep architectures. Our construction provides a way of reasoning about double descent that allows us to factor in the ill-conditioning of the learning problem. Rather than focusing simply on the model size, it suggests that for neural networks the quantity of interest might also be $\widehat{\lambda}_{\min}^+$ for intermediary features, which is affected by size of the model but also by the distribution of the weights and architectural choices. For now we present more empirical explorations and ablations in Appendix G, and put forward this perspective as a conjecture for further exploration.

## 7 Conclusion and Future Work

In this work we analyse the double descent phenomenon in the context of the least squares problem. We make the observation that the excess risk of gradient descent is controlled by the smallest *positive* eigenvalue, $\widehat{\lambda}_{\min}^+$, of the feature covariance matrix. Furthermore, this quantity follows the Bai-Yin law with high probability under mild distributional assumptions on features, that is, it manifests a U-shaped behaviour as the number of features increases, which we argue induces a double descent shape of the excess risk. Through this we provide a connection between the widely known phenomenon and optimization process and conditioning of the problem. We believe this insight provides a different perspective compared to existing results focusing on the Moore-Penrose pseudo-inverse solution.

While we do not show the same formally for the more complex learning scenario of neural networks, our work conjectures the connection between the known double descent shape and model size is through $\widehat{\lambda}_{\min}^+$ of the features extracted from the intermediary layers. For the least squares problem $\widehat{\lambda}_{\min}^+$ correlates strongly with model size (and hence feature size). However this might not necessarily be always true for neural networks. For example we show empirically that while both depth and width increase the model size, they might affect $\widehat{\lambda}_{\min}^+$ differently. We believe that our work could enable much needed effort, either empirical or theoretical, to disentangle further the role of various factors, like depth and width or other architectural choices like skip connections on double descent.

## Acknowledgements

We thank the anonymous reviewers for their helpful comments.

Csaba Szepesvári gratefully acknowledges funding from the Canada CIFAR AI Chairs Program, the Natural Sciences and Engineering Research Council (NSERC) of Canada, and the Alberta Machine Intelligence Institute (Amii).

Omar Rivasplata gratefully acknowledges funding from the U.K. Engineering and Physical Sciences Research Council (EPSRC) under grant number EP/W522636/1, and sponsorship from DeepMind. This work was done while Omar was a research scientist intern at DeepMind.

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
