# A  Definitions from Linear Analysis

We denote column vectors and matrices with small and capital bold letters, respectively, e.g. $\boldsymbol{\alpha} = [\alpha_1, \alpha_2, \ldots, \alpha_d]^\top \in \mathbb{R}^d$ and $\boldsymbol{A} \in \mathbb{R}^{d_1 \times d_2}$. The singular values of a rectangular matrix $\boldsymbol{A} \in \mathbb{R}^{n \times d}$ are non-negative and are denoted $s_{\max}(\boldsymbol{A}) = s_1(\boldsymbol{A}) \geq \ldots \geq s_{n \wedge d}(\boldsymbol{A}) = s_{\min}(\boldsymbol{A})$. The rank of $\boldsymbol{A}$ is $r = \max\{k \mid s_k(\boldsymbol{A}) > 0\}$. The eigenvalues of a Positive Semi-Definite (PSD) matrix $\boldsymbol{M} \in \mathbb{R}^{d \times d}$ are non-negative and are denoted $\lambda_{\max}(\boldsymbol{M}) = \lambda_1(\boldsymbol{M}) \geq \ldots \geq \lambda_d(\boldsymbol{M}) = \lambda_{\min}(\boldsymbol{M})$, while $\lambda_{\min}^+(\boldsymbol{M})$ denotes the smallest *positive* (*non-zero*) eigenvalue.

When $\boldsymbol{M} \in \mathbb{R}^{d \times d}$ is positive definite, we define $\|\boldsymbol{x}\|_{\boldsymbol{M}}$ for $\boldsymbol{x} \in \mathbb{R}^d$ by $\|\boldsymbol{x}\|_{\boldsymbol{M}} = \sqrt{\boldsymbol{x}^\top \boldsymbol{M} \boldsymbol{x}}$ . It is easy to check that $\| \cdot \|_{\boldsymbol{M}}$ is indeed a norm on $\mathbb{R}^d$, hence it induces a metric over $\mathbb{R}^d$, with the distance between $\boldsymbol{x}$ and $\boldsymbol{y}$ given by $\|\boldsymbol{x} - \boldsymbol{y}\|_{\boldsymbol{M}} = \sqrt{(\boldsymbol{x} - \boldsymbol{y})^\top \boldsymbol{M} (\boldsymbol{x} - \boldsymbol{y})}$. If $\boldsymbol{M}$ is only semi-definite, these definitions would give a semi-norm and semi-metric. Note that $\|\boldsymbol{x}\|_{\boldsymbol{M}} = \|\boldsymbol{M}^{1/2} \boldsymbol{x}\|$ where $\boldsymbol{M}^{1/2}$ is the matrix square root of $\boldsymbol{M}$. If we set $\boldsymbol{M} = \boldsymbol{I}$, the identity matrix, then the norm $\| \cdot \|_{\boldsymbol{M}}$ reduces to the standard Euclidean norm: $\|\boldsymbol{x}\| = \sqrt{\boldsymbol{x}^\top \boldsymbol{x}}$.

Combining the Cauchy-Schwarz inequality and the definition of operator norm $\|\boldsymbol{M}\| = s_{\max}(\boldsymbol{M})$, which implies $\|\boldsymbol{M}\boldsymbol{x}\| \leq \|\boldsymbol{M}\|\|\boldsymbol{x}\|$, we get the inequality $\|\boldsymbol{x}\|_{\boldsymbol{M}}^2 \leq \|\boldsymbol{x}\|^2 \|\boldsymbol{M}\|$.

# B  Regression Model

Consider a linear regression model

$$Y = \boldsymbol{X}^\top \boldsymbol{w}^\star + \varepsilon \,,$$

where $\boldsymbol{w}^\star \in \mathbb{R}^d$ is fixed and unknown, the random input $\boldsymbol{X} \in \mathbb{R}^d$ is distributed according to some unknown distribution $P_X$ supported on a unit ball, and noise is given by a random variable $\varepsilon$ independent from $\boldsymbol{X}$, such that $\mathbb{E}[\varepsilon] = 0$ and $\mathbb{E}[\varepsilon^2] = \sigma^2$. After observing an i.i.d. training sample $S = ((\boldsymbol{X}_i, Y_i))_{i=1}^n$, we run GD on the empirical squared loss:

$$\hat{L}_S(\boldsymbol{w}) = \frac{1}{2n} \sum_{i=1}^n (\boldsymbol{w}^\top \boldsymbol{X}_i - Y_i)^2 \,.$$

The sample covariance matrix is $\widehat{\boldsymbol{\Sigma}} = (\boldsymbol{X}_1 \boldsymbol{X}_1^\top + \cdots + \boldsymbol{X}_n \boldsymbol{X}_n^\top)/n$, and its population counterpart is the covariance matrix $\boldsymbol{\Sigma} = \mathbb{E}[\boldsymbol{X}\boldsymbol{X}^\top]$. Let $\widehat{\boldsymbol{\Sigma}} = \boldsymbol{U}\boldsymbol{S}\boldsymbol{V}^\top$ be the SVD of $\widehat{\boldsymbol{\Sigma}}$ with orthogonal matrices $\boldsymbol{U} = [\boldsymbol{u}_1, \ldots, \boldsymbol{u}_d]$ and $\boldsymbol{V} = [\boldsymbol{v}_1, \ldots, \boldsymbol{v}_d]$, and scaling matrix $\boldsymbol{S} = \mathrm{diag}(s_1(\widehat{\boldsymbol{\Sigma}}), \ldots, s_d(\widehat{\boldsymbol{\Sigma}}))$ of singular values arranged in decreasing order: $s_1(\widehat{\boldsymbol{\Sigma}}) \geq s_2(\widehat{\boldsymbol{\Sigma}}) \geq \cdots \geq s_d(\widehat{\boldsymbol{\Sigma}}) \geq 0$. Note that $s_i(\widehat{\boldsymbol{\Sigma}}) = \lambda_i(\widehat{\boldsymbol{\Sigma}}) =: \widehat{\lambda}_i$ since $\widehat{\boldsymbol{\Sigma}}$ is positive semi-definite. The matrix $\widehat{\boldsymbol{\Sigma}}$ might be degenerate ($\widehat{\lambda}_d = 0$), and the non-degenerate part is given by $\boldsymbol{U}_r := [\boldsymbol{u}_1, \ldots, \boldsymbol{u}_r]$, $\boldsymbol{V}_r := [\boldsymbol{v}_1, \ldots, \boldsymbol{v}_r]$ and $\boldsymbol{S}_r := \mathrm{diag}(\widehat{\lambda}_1, \ldots, \widehat{\lambda}_r)$, where $r = \mathrm{rank}(\widehat{\boldsymbol{\Sigma}})$. We denote $\widehat{\boldsymbol{M}} = \boldsymbol{U}_r \boldsymbol{U}_r^\top$, and note that $\widehat{\boldsymbol{M}}^2 = \widehat{\boldsymbol{M}}$. For the minimal *positive* (*non-zero*) eigenvalue we use the shorthand $\widehat{\lambda}_{\min}^+ = \lambda_{\min}^+(\widehat{\boldsymbol{\Sigma}}) = \lambda_r(\widehat{\boldsymbol{\Sigma}})$.

# C  Proof of Theorem 2 (Excess Risk of GD)

In this section we consider the standard GD algorithm, that is $\mathcal{A}_S(\boldsymbol{w}_0) = \boldsymbol{w}_T$, which is obtained recursively by applying the update rule $\boldsymbol{w}_{t+1} = \boldsymbol{w}_t - \alpha \nabla \hat{L}_S(\boldsymbol{w}_t)$ with some step size $\alpha > 0$ and initialization $\boldsymbol{w}_0 \in \mathbb{R}^d$. The rule is iterated for $t = 0, \ldots, T-1$.

Recall that the *excess risk* of $\mathcal{A}_S(\boldsymbol{w}_0)$ is defined as

$$\mathcal{E}_T = L(\mathcal{A}_S(\boldsymbol{w}_0)) - L(\boldsymbol{w}^\star) \,.$$

Next we give upper bounds on the excess risk of GD output.

**Theorem 2 (restated).** *Assume that $\alpha \leq 1/\widehat{\lambda}_1$. For any $\boldsymbol{w}_0 \in \mathbb{R}^d$ and $x > 0$, with probability at least $1 - e^{-x}$ over inputs we have*

$$\mathbb{E}_{\boldsymbol{\varepsilon}}[\mathcal{E}_T] \leq \widehat{\lambda}_1^2 (1 - \alpha\widehat{\lambda}_{\min}^+)^{2T} \|\boldsymbol{w}^\star - \boldsymbol{w}_0\|^2 + \frac{4\sigma^2}{n}\left(\frac{\widehat{\lambda}_1}{\widehat{\lambda}_{\min}^+}\right)^2$$
$$+ \frac{12}{\sqrt{n}}\left(\sqrt{\ln d} + \sqrt{x}\right)\left(\|\boldsymbol{w}_0\|^2 + 2\|\boldsymbol{w}^\star\|^2 + \sigma^2 \sum_{i=1}^r \widehat{\lambda}_i^{-1}\right).$$

The proof of Theorem 2 is based on a the following decomposition, and remaining subsections will be dedicated to bounding the constituent terms.

**Proposition 1 (restated).** *For any $\boldsymbol{w}_0 \in \mathbb{R}^d$,*

$$\mathcal{E}_T \leq \underbrace{\|\mathcal{A}_S(\boldsymbol{w}_0) - \mathcal{A}_S(\boldsymbol{w}^\star)\|_{\widehat{\boldsymbol{\Sigma}}}^2}_{(1)} + \underbrace{\|\mathcal{A}_S(\boldsymbol{w}^\star) - \boldsymbol{w}^\star\|_{\widehat{\boldsymbol{\Sigma}}}^2}_{(2)} + \underbrace{\|\boldsymbol{\Sigma} - \widehat{\boldsymbol{\Sigma}}\|_2\left(\|\mathcal{A}_S(\boldsymbol{w}_0)\|^2 + \|\boldsymbol{w}^\star\|^2\right)}_{(3)}.$$

*Proof.* Observe that for the square loss we have

$$L(\mathcal{A}_S(\boldsymbol{w}_0)) - L(\boldsymbol{w}^\star) = \frac{1}{2}\|\mathcal{A}_S(\boldsymbol{w}_0) - \boldsymbol{w}^\star\|_{\boldsymbol{\Sigma}}^2$$
$$= \frac{1}{2}\|\mathcal{A}_S(\boldsymbol{w}_0) - \boldsymbol{w}^\star\|_{\widehat{\boldsymbol{\Sigma}}}^2 + \frac{1}{2}\|\mathcal{A}_S(\boldsymbol{w}_0) - \boldsymbol{w}^\star\|_{\boldsymbol{\Sigma}-\widehat{\boldsymbol{\Sigma}}}^2$$
$$= \frac{1}{2}\underbrace{\|\mathcal{A}_S(\boldsymbol{w}_0) - \boldsymbol{w}^\star\|_{\widehat{\boldsymbol{\Sigma}}}^2}_{(a)} + \frac{1}{2}\underbrace{\|\mathcal{A}_S(\boldsymbol{w}_0) - \boldsymbol{w}^\star\|_{\boldsymbol{\Sigma}-\widehat{\boldsymbol{\Sigma}}}^2}_{(b)}.$$

Note that bounding term $(a)$ reduces to

$$\|\mathcal{A}_S(\boldsymbol{w}_0) - \boldsymbol{w}^\star\|_{\widehat{\boldsymbol{M}}}^2 \leq 2\|\mathcal{A}_S(\boldsymbol{w}_0) - \mathcal{A}_S(\boldsymbol{w}^\star)\|_{\widehat{\boldsymbol{\Sigma}}}^2 + 2\|\mathcal{A}_S(\boldsymbol{w}^\star) - \boldsymbol{w}^\star\|_{\widehat{\boldsymbol{\Sigma}}}^2.$$

On the other we handle term $(b)$ by Cauchy-Schwarz and triangle inequalities:

$$\|\mathcal{A}_S(\boldsymbol{w}_0) - \boldsymbol{w}^\star\|_{\boldsymbol{\Sigma}-\widehat{\boldsymbol{\Sigma}}}^2 \leq 2\|\boldsymbol{\Sigma} - \widehat{\boldsymbol{\Sigma}}\|\left(\|\mathcal{A}_S(\boldsymbol{w}_0)\|^2 + \|\boldsymbol{w}^\star\|^2\right).$$

$\square$

Here, the first term $(1)$, vanishes as long as the algorithm-map $\mathcal{A}_S$ is contractive on the aforementioned subspace, which we show in Appendix C.1 thanks to the closed-form expression of GD iterates.

Term $(2)$ captures algorithm's sensitivity to the label noise: How far would GD go when initialized at the *minimum of the risk*? Indeed it is easy to see that when there is no label noise, term $(2)$ is zero (one can demonstrate it by the descent lemma). In the presence of noise, matters are more complicated, and we employ more or less standard technique where we recursively track the distance between GD iterates and "virtual" iterates, which are obtained as if we could remove the noise. This results in a bound $4(\sigma^2/n)(\widehat{\lambda}_{\min}^+)^{-2}$, which is shown in Appendix C.3.

Finally, term $(3)$ is essentially a concentration of the sample covariance matrix and ensuring that the norm of the solution remains well-behaved, see Appendix C.2. The concentration of the sample covariance matrix is due to the matrix Chernoff inequality [Tropp, 2012]. We control the norm of the solution by relating it to the Moore-Penrose pseudoinverse solution which can be written in a closed-form. This makes it easy to see that when the label noise is absent, the norm depends only on $\boldsymbol{w}^\star$ and $\boldsymbol{w}_0$. On the other hand, when the noise is present things will depend on behaviour of $\widehat{\lambda}_{\min}^+$ as discussed in Section 3.2.

### C.1 Contractivity of GD (Term (1))

Contractivity of GD comes from the following straightforward proposition.

**Proposition 2.** *For a $T$-step gradient descent map $\mathcal{A}_S : \mathbb{R}^d \to \mathbb{R}^d$ with step size $\alpha > 0$ applied to the least squares, and for all $\boldsymbol{w}_0 \in \mathbb{R}^d$, we have a.s. that*

$$\mathcal{A}_S(\boldsymbol{w}_0) = (\boldsymbol{I} - \alpha\widehat{\boldsymbol{\Sigma}})^T \boldsymbol{w}_0 + \alpha \sum_{t=0}^{T-1} (\boldsymbol{I} - \alpha\widehat{\boldsymbol{\Sigma}})^t \left( \frac{1}{n} \sum_{i=1}^{n} \boldsymbol{X}_i Y_i \right) .$$

*Proof.* Abbreviate $\boldsymbol{C} = (\boldsymbol{X}_1 Y_1 + \cdots + \boldsymbol{X}_n Y_n)/n$. Since $\nabla \widehat{L}_S(\boldsymbol{w}) = \widehat{\boldsymbol{\Sigma}}\boldsymbol{w} - \boldsymbol{C}$, observe that

$$\boldsymbol{w}_t = \boldsymbol{w}_{t-1} - \alpha(\widehat{\boldsymbol{\Sigma}}\boldsymbol{w}_{t-1} - \boldsymbol{C}) = (\boldsymbol{I} - \alpha\widehat{\boldsymbol{\Sigma}})\boldsymbol{w}_{t-1} + \alpha\boldsymbol{C} .$$

A simple recursive argument reveals that for every $\boldsymbol{w}_0 \in \mathbb{R}^d$

$$\begin{aligned}
\mathcal{A}_S(\boldsymbol{w}_0) = \boldsymbol{w}_T &= (\boldsymbol{I} - \alpha\widehat{\boldsymbol{\Sigma}})\boldsymbol{w}_{T-1} + \alpha\boldsymbol{C} \\
&= (\boldsymbol{I} - \alpha\widehat{\boldsymbol{\Sigma}})^2 \boldsymbol{w}_{T-2} + \alpha(\boldsymbol{I} - \alpha\widehat{\boldsymbol{\Sigma}})\boldsymbol{C} + \alpha\boldsymbol{C} \\
&= (\boldsymbol{I} - \alpha\widehat{\boldsymbol{\Sigma}})^3 \boldsymbol{w}_{T-3} + \alpha(\boldsymbol{I} - \alpha\widehat{\boldsymbol{\Sigma}})^2\boldsymbol{C} + \alpha(\boldsymbol{I} - \alpha\widehat{\boldsymbol{\Sigma}})\boldsymbol{C} + \alpha\boldsymbol{C} \\
&\cdots \\
&= (\boldsymbol{I} - \alpha\widehat{\boldsymbol{\Sigma}})^T \boldsymbol{w}_0 + \alpha \sum_{t=0}^{T-1} (\boldsymbol{I} - \alpha\widehat{\boldsymbol{\Sigma}})^t \boldsymbol{C} .
\end{aligned}$$

$\square$

Proposition 2 implies the following simple fact.

**Corollary 1** (Contractivity of GD). *The $T$-step gradient descent map $\mathcal{A}_S : \mathbb{R}^d \to \mathbb{R}^d$ with step size $\alpha > 0$ applied to the least squares problem satisfies, for all $\boldsymbol{w}_0, \boldsymbol{u}_0 \in \mathbb{R}^d$,*

$$\|\mathcal{A}_S(\boldsymbol{w}_0) - \mathcal{A}_S(\boldsymbol{u}_0)\|_{\widehat{\boldsymbol{\Sigma}}} \leq \widehat{\lambda}_1 (1 - \alpha\widehat{\lambda}_{\min}^+)^T \|\boldsymbol{w}_0 - \boldsymbol{u}_0\| .$$

*Proof.* Clearly $\|\mathcal{A}_S(\boldsymbol{w}_0) - \mathcal{A}_S(\boldsymbol{u}_0)\|_{\widehat{\boldsymbol{\Sigma}}} \leq \widehat{\lambda}_1 \|\mathcal{A}_S(\boldsymbol{w}_0) - \mathcal{A}_S(\boldsymbol{u}_0)\|_{\boldsymbol{U}_r \boldsymbol{U}_r^\top}$. By Proposition 2 for any $\boldsymbol{w}_0, \boldsymbol{u}_0 \in \mathbb{R}^d$:

$$\begin{aligned}
\|\mathcal{A}_S(\boldsymbol{w}_0) - \mathcal{A}_S(\boldsymbol{u}_0)\|_{\boldsymbol{U}_r \boldsymbol{U}_r^\top} &= \|(\boldsymbol{I} - \alpha\widehat{\boldsymbol{\Sigma}})^T(\boldsymbol{w}_0 - \boldsymbol{u}_0)\|_{\boldsymbol{U}_r \boldsymbol{U}_r^\top} \\
&= \|\boldsymbol{U}_r^\top (\boldsymbol{I} - \alpha\widehat{\boldsymbol{\Sigma}})^T(\boldsymbol{w}_0 - \boldsymbol{u}_0)\| \\
&\leq \|\boldsymbol{U}_r^\top (\boldsymbol{I} - \alpha\widehat{\boldsymbol{\Sigma}})^T\| \|(\boldsymbol{w}_0 - \boldsymbol{u}_0)\| .
\end{aligned}$$

Now,

$$\boldsymbol{U}_r^\top (\boldsymbol{I} - \alpha\widehat{\boldsymbol{\Sigma}})^T = \boldsymbol{U}_r^\top \boldsymbol{U} (\boldsymbol{I} - \alpha\boldsymbol{S})^T \boldsymbol{V}^\top = \boldsymbol{I}_{r \times d}(\boldsymbol{I} - \alpha\boldsymbol{S})^T \boldsymbol{V}^\top = (\boldsymbol{I}_{r \times d} - \alpha\boldsymbol{S}_{r \times d})^T \boldsymbol{V}^\top$$

where subscript $r \times n$ stands for clipping the matrix to $r$ rows and $d$ columns. The above implies that the operator norm of $\boldsymbol{U}_r^\top (\boldsymbol{I} - \alpha\widehat{\boldsymbol{\Sigma}})^T$ satisfies $\|\boldsymbol{U}_r^\top (\boldsymbol{I} - \alpha\widehat{\boldsymbol{\Sigma}})^T\|_2 \leq (1 - \alpha\lambda_{\min}^+(\widehat{\boldsymbol{\Sigma}}))^T$. $\square$

### C.2  Concentration of Spectrum (Term (3))

Our next goal is to understand the behaviour of

$$\|\boldsymbol{\Sigma} - \widehat{\boldsymbol{\Sigma}}\|_2 \left( \|\mathcal{A}_S(\boldsymbol{w}_0)\|^2 + \|\boldsymbol{w}^\star\|^2 \right) . \tag{2}$$

A high-probability concentration of a covariance matrix is readily given by the matrix Chernoff inequality:

**Theorem 3** (Tropp [2012]). *Suppose that inputs have a bounded spectral norm which is at most $B_X$. Then,*

$$\mathbb{P}\left( \|\widehat{\boldsymbol{\Sigma}} - \boldsymbol{\Sigma}\|_2 \geq \frac{6B_X}{\sqrt{n}} \left( \sqrt{\ln d} + \sqrt{x} \right) \right) \leq e^{-x} .$$

Next we need to control $\|\mathcal{A}_S(\boldsymbol{w}_0)\|^2$. This will be done by using a closed-form of the GD iterate at step $T$ given by Proposition 2:

$$\|\mathcal{A}_S(\boldsymbol{w}_0)\|^2 \leq 2 \underbrace{\|(\boldsymbol{I} - \alpha\widehat{\boldsymbol{\Sigma}})^T \boldsymbol{w}_0\|_2^2}_{(a)} + 2 \left\| \underbrace{\alpha \sum_{t=0}^{T-1}(\boldsymbol{I} - \alpha\widehat{\boldsymbol{\Sigma}})^t \left(\frac{1}{n}\sum_{i=1}^n \boldsymbol{X}_i Y_i\right)}_{(b)} \right\|_2^2$$

where Cauchy-Schwarz inequality gives $(a) \leq \|\boldsymbol{w}_0\|^2$, while $(b)$ converges to the Moore-Penrose pseudoinverse of $\widehat{\boldsymbol{\Sigma}}$ as $T \to \infty$ as long as $\alpha \leq 1/\widehat{\lambda}_1$ [Ben-Israel and Charnes, 1963]. Hence, we will need to quantify its squared $\ell_2$ norm:

**Proposition 3.** *Suppose that given some* $\boldsymbol{w}^\star \in \mathbb{R}^d$, $\boldsymbol{D}^\top = [\boldsymbol{x}_i, \ldots, \boldsymbol{x}_n] \in \mathbb{R}^{d\times n}$, *and* $\boldsymbol{\varepsilon} \in \mathbb{R}^n$, *we have* $\boldsymbol{y} = \boldsymbol{D}\boldsymbol{w}^\star + \boldsymbol{\varepsilon}$. *Then, the Moore-Penrose pseudoinverse solution* $\boldsymbol{w}^{\mathrm{pinv}} = (\boldsymbol{D}^\top\boldsymbol{D})^\dagger \boldsymbol{D}^\top \boldsymbol{y}$ *satisfies*

$$\|\boldsymbol{w}^{\mathrm{pinv}}\|^2 = \|\boldsymbol{w}^\star\|^2 + 2\boldsymbol{\varepsilon}^\top(\boldsymbol{D}^\top\boldsymbol{D})^\dagger\boldsymbol{w}^\star + \|\boldsymbol{\varepsilon}\|_{(\boldsymbol{D}\boldsymbol{D}^\top)^{-1}}^2 .$$

*Proof.* Observe that

$$\begin{aligned}
\|\boldsymbol{w}^{\mathrm{pinv}}\|^2 &= \boldsymbol{y}^\top\boldsymbol{D}(\boldsymbol{D}^\top\boldsymbol{D})^{\dagger 2}\boldsymbol{D}^\top\boldsymbol{y}\\
&= (\boldsymbol{D}\boldsymbol{w}^\star + \boldsymbol{\varepsilon})^\top\boldsymbol{D}(\boldsymbol{D}^\top\boldsymbol{D})^{\dagger 2}\boldsymbol{D}^\top(\boldsymbol{D}\boldsymbol{w}^\star + \boldsymbol{\varepsilon})\\
&= \|\boldsymbol{w}^\star\|^2 + 2\boldsymbol{\varepsilon}^\top(\boldsymbol{D}^\top\boldsymbol{D})^{\dagger 2}\boldsymbol{D}^\top\boldsymbol{D}\boldsymbol{w}^\star + \boldsymbol{\varepsilon}^\top\boldsymbol{D}(\boldsymbol{D}^\top\boldsymbol{D})^{\dagger 2}\boldsymbol{D}^\top\boldsymbol{\varepsilon}\\
&= \|\boldsymbol{w}^\star\|^2 + 2\boldsymbol{\varepsilon}^\top(\boldsymbol{D}^\top\boldsymbol{D})^\dagger\boldsymbol{w}^\star + \boldsymbol{\varepsilon}^\top(\boldsymbol{D}\boldsymbol{D}^\top)^{-1}\boldsymbol{\varepsilon} .
\end{aligned}$$

$\square$

Putting things together we have that w.p. at least $1 - e^{-x}$ over $(\boldsymbol{X}_1, \ldots, \boldsymbol{X}_n)$,

$$\text{Eq. (2)} \leq \frac{12}{\sqrt{n}}\left(\sqrt{\ln d} + \sqrt{x}\right)\left(\|\boldsymbol{w}_0\|^2 + 2\|\boldsymbol{w}^\star\|^2 + \frac{2}{n}\boldsymbol{\varepsilon}^\top\widehat{\boldsymbol{\Sigma}}^\dagger\boldsymbol{w}^\star + \|\boldsymbol{\varepsilon}\|_{(\boldsymbol{D}\boldsymbol{D}^\top)^{-1}}^2\right) .$$

Moreover, note that taking expectation over label noise gives

$$\mathbb{E}_{\boldsymbol{\varepsilon}}[\text{Eq. (2)}] \leq \frac{12}{\sqrt{n}}\left(\sqrt{\ln d} + \sqrt{x}\right)\left(\|\boldsymbol{w}_0\|^2 + 2\|\boldsymbol{w}^\star\|^2 + \sigma^2\sum_{i=1}^r \widehat{\lambda}_i^{-1}\right) .$$

### C.3 Bounding term (2)

The goal of this section is to bound

$$\|\mathcal{A}_S(\boldsymbol{w}^\star) - \boldsymbol{w}^\star\|_{\widehat{\boldsymbol{\Sigma}}}^2 ,$$

which captures how sensitive the algorithm is to the noise when initialize at the optimum. When there is no label noise, the standard descent lemma (not shown here) readily gives that the term vanishes.

**Lemma 2** (Descent Lemma). *Assuming that* $\alpha \leq 1/\widehat{\lambda}_1$,

$$\sum_{t=0}^{T-1}\|\nabla\hat{L}_S(\boldsymbol{w}_t)\|^2 \leq \frac{2}{\alpha}\left(\hat{L}_S(\boldsymbol{w}_0) - \hat{L}_S(\mathcal{A}_S(\boldsymbol{w}_0))\right) .$$

In particular, if $\boldsymbol{w}_t^\star$ are the iterates of GD when starting from $\boldsymbol{w}^\star$ (so that $\boldsymbol{w}_0 = \boldsymbol{w}^\star$), then

$$\begin{aligned}
\|\mathcal{A}_S(\boldsymbol{w}^\star) - \boldsymbol{w}^\star\|_{\boldsymbol{M}}^2 &= \left\|\alpha\sum_{t=0}^{T-1}\nabla\hat{L}_S(\boldsymbol{w}_t^\star)\right\|_{\boldsymbol{M}}^2\\
&\leq \alpha^2 T\sum_{t=0}^{T-1}\|\nabla\hat{L}_S(\boldsymbol{w}_t^\star)\|_{\boldsymbol{M}}^2\\
&\leq \alpha^2\frac{2T}{\alpha}\left(\hat{L}_S(\boldsymbol{w}^\star) - \hat{L}_S(\boldsymbol{w}_T^\star)\right)\\
&\leq 2\alpha T\hat{L}_S(\boldsymbol{w}^\star) = 0 .
\end{aligned}$$

Here we consider the case with i.i.d. label noise $(\varepsilon_1, \ldots, \varepsilon_n)$ and therefore show a more general handling of the term. Recall that $\mathbb{E}[\varepsilon_i] = 0$ and $\mathbb{E}[\varepsilon_i^2] = \sigma^2$ for $i \in [n]$. Throughout this section abbreviate $\mathbb{E}[\cdot \mid \boldsymbol{X}_1, \ldots, \boldsymbol{X}_n] = \mathbb{E}_{\boldsymbol{\varepsilon}}[\cdot]$.

**Lemma 3.** *Let $\widehat{\boldsymbol{M}}$ be defined as in Section 3.1. For any $T > 0$, GD achieves*

$$\mathbb{E}_{\boldsymbol{\varepsilon}}\left[\|\boldsymbol{w}^\star - \mathcal{A}_S(\boldsymbol{w}^\star)\|_{\widehat{\boldsymbol{\Sigma}}}^2\right] \leq \frac{4\sigma^2}{n}\left(\frac{\widehat{\lambda}_1}{\widehat{\lambda}_{\min}^+}\right)^2.$$

*Proof.* We begin by noting that the integral form of Taylor theorem gives us that for any $\boldsymbol{w}^\star \in \arg\min_{\boldsymbol{w} \in \mathbb{R}^d} \hat{L}(\boldsymbol{w})$ and any $\boldsymbol{w} \in \mathbb{R}^d$,

$$\hat{L}(\boldsymbol{w}) - \hat{L}(\boldsymbol{w}^\star) = \frac{1}{2}(\boldsymbol{w} - \boldsymbol{w}^\star)^\top \left(\int_0^1 \nabla^2 \hat{L}(\tau\boldsymbol{w} + (1-\tau)\boldsymbol{w}^\star)\,\mathrm{d}\tau\right)(\boldsymbol{w} - \boldsymbol{w}^\star)$$

$$\geq \frac{1}{2} \cdot \widehat{\lambda}_{\min}^+ (\boldsymbol{w} - \boldsymbol{w}^\star)^\top \widehat{\boldsymbol{M}}(\boldsymbol{w} - \boldsymbol{w}^\star).$$

Thus, taking $\boldsymbol{w} = \mathcal{A}_S(\boldsymbol{w}^\star)$, we have

$$\mathbb{E}_{\boldsymbol{\varepsilon}}\left[\|\boldsymbol{w}^\star - \mathcal{A}_S(\boldsymbol{w}^\star)\|_{\widehat{\boldsymbol{M}}}^2\right] \leq \frac{1}{\widehat{\lambda}_{\min}^+}\left(\mathbb{E}_{\boldsymbol{\varepsilon}}\,\hat{L}(\boldsymbol{w}^\star) - \mathbb{E}_{\boldsymbol{\varepsilon}}\left[\hat{L}(\mathcal{A}_S(\boldsymbol{w}^\star))\right]\right)$$

$$= \frac{1}{\widehat{\lambda}_{\min}^+}\left(\sigma^2 - \mathbb{E}_{\boldsymbol{\varepsilon}}\left[\hat{L}(\mathcal{A}_S(\boldsymbol{w}^\star))\right]\right).$$

Now, let's focus on the loss term on the r.h.s.:

$$\mathbb{E}_{\boldsymbol{\varepsilon}}\left[\hat{L}_S(\boldsymbol{w}_T^\star)\right] = \frac{1}{n}\sum_{i=1}^n \mathbb{E}_{\boldsymbol{\varepsilon}}\left[\left((\boldsymbol{w}_T^\star - \boldsymbol{w}_0^\star)^\top \boldsymbol{X}_i - \varepsilon_i\right)^2\right]$$

$$= \sigma^2 - \frac{2}{n}\sum_{i=1}^n \mathbb{E}_{\boldsymbol{\varepsilon}}\left[\varepsilon_i\,(\boldsymbol{w}_T^\star - \boldsymbol{w}_0^\star)^\top \boldsymbol{X}_i\right] + \mathbb{E}_{\boldsymbol{\varepsilon}}\left[(\boldsymbol{w}_T^\star - \boldsymbol{w}_0^\star)^\top \widehat{\boldsymbol{\Sigma}}\,(\boldsymbol{w}_T^\star - \boldsymbol{w}_0^\star)\right]$$

$$\geq \sigma^2 - \frac{2}{n}\sum_{i=1}^n \mathbb{E}_{\boldsymbol{\varepsilon}}\left[\varepsilon_i\,(\boldsymbol{w}_T^\star - \boldsymbol{w}_0^\star)^\top \boldsymbol{X}_i\right]$$

$$= \sigma^2 - \frac{2}{n}\sum_{i=1}^n \mathbb{E}_{\boldsymbol{\varepsilon}}\left[\varepsilon_i\,\boldsymbol{w}_T^{\star\top}\boldsymbol{X}_i\right]$$

where the last term is small when label noise is not too correlated with the output $\boldsymbol{w}_T^\star$. Hence to control the term, we need to measure the effect of the noise on GD. To do so we will introduce an additional iterates $(\tilde{\boldsymbol{w}}_t)_t$ constructed by running GD on labels without noise, that is

$$\tilde{\boldsymbol{w}}_{t+1}^\star = \tilde{\boldsymbol{w}}_t^\star - \alpha\nabla\tilde{L}_S(\tilde{\boldsymbol{w}}_t^\star) \qquad \text{where} \quad \tilde{L}(\boldsymbol{w}) = \frac{1}{2n}\sum_{i=1}^n \left(\boldsymbol{w}^\top \boldsymbol{X}_i - \boldsymbol{w}^{\star\top}\boldsymbol{X}_i\right)^2.$$

The plan is then to bound the deviation $\|\boldsymbol{w}_T^\star - \tilde{\boldsymbol{w}}_T^\star\|_{\widehat{\boldsymbol{M}}}$ which we will do recursively. We proceed:

$$\frac{2}{n}\sum_{i=1}^n \mathbb{E}_{\boldsymbol{\varepsilon}}\left[\varepsilon_i\,\boldsymbol{w}_T^{\star\top}\boldsymbol{X}_i\right]$$

$$= \frac{2}{n}\sum_{i=1}^n \mathbb{E}_{\boldsymbol{\varepsilon}}\left[\varepsilon_i(\boldsymbol{w}_T^\star - \tilde{\boldsymbol{w}}_T^\star)^\top \boldsymbol{X}_i\right] \qquad\qquad \text{(Note that } \mathbb{E}_{\boldsymbol{\varepsilon}}[\tilde{\boldsymbol{w}}_T^\star \mid \boldsymbol{X}_i] = 0)$$

$$= \frac{2}{n}\sum_{i=1}^n \mathbb{E}_{\boldsymbol{\varepsilon}}\left[\varepsilon_i(\boldsymbol{w}_T^\star - \tilde{\boldsymbol{w}}_T^\star)^\top \widehat{\boldsymbol{M}}\boldsymbol{X}_i\right] \qquad\qquad \text{(Since } \widehat{\boldsymbol{M}}\boldsymbol{X}_i = \boldsymbol{X}_i)$$

$$\leq \frac{2}{n}\mathbb{E}_{\boldsymbol{\varepsilon}}\left[\left\|\sum_{i=1}^n \varepsilon_i\boldsymbol{X}_i\right\|\left\|\widehat{\boldsymbol{M}}(\boldsymbol{w}_T^\star - \tilde{\boldsymbol{w}}_T^\star)\right\|\right] \qquad\qquad \text{(Cauchy-Schwarz)}$$

Now we will handle $\left\|\widehat{M}(w_T^\star - \tilde{w}_T^\star)\right\| = \|w_T^\star - \tilde{w}_T^\star\|_{\widehat{M}}$ by following a recursive argument. First, observe that for any $t = 0, 1, 2, \ldots$

$$\nabla \hat{L}(\tilde{w}_t^\star) = \widehat{\Sigma}\tilde{w}_t^\star - \frac{1}{n}\sum_{i=1}^n X_i X_i^\top w_0^\star = \widehat{\Sigma}(\tilde{w}_t^\star - w_0^\star),$$

and at the same time

$$\nabla \hat{L}(w_t^\star) = \widehat{\Sigma}w_t^\star - \frac{1}{n}\sum_{i=1}^n X_i X_i^\top w_0^\star - \frac{1}{n}\sum_{i=1}^n X_i\varepsilon_i = \widehat{\Sigma}(w_t^\star - w_0^\star) - \frac{1}{n}\sum_{i=1}^n X_i\varepsilon_i.$$

Thus,

$$\|w_{t+1}^\star - \tilde{w}_{t+1}^\star\|_{\widehat{M}} = \left\|w_t^\star - \tilde{w}_t^\star - \alpha\left(\nabla\hat{L}(w_t^\star) - \nabla\hat{L}(\tilde{w}_t^\star)\right)\right\|_{\widehat{M}} \tag{3}$$

$$= \left\|w_t^\star - \tilde{w}_t^\star - \alpha\widehat{\Sigma}(w_t^\star - \tilde{w}_t^\star) - \frac{\alpha}{n}\sum_{i=1}^n X_i\varepsilon_i\right\|_{\widehat{M}}$$

$$= \left\|(I - \alpha\widehat{\Sigma})(w_t^\star - \tilde{w}_t^\star)\right\|_{\widehat{M}} + \frac{\alpha}{n}\left\|\sum_{i=1}^n X_i\varepsilon_i\right\|_{\widehat{M}}$$

$$\overset{(a)}{\leq} \|I - \alpha\widehat{\Sigma}\|_{\widehat{M}}\|w_t^\star - \tilde{w}_t^\star\|_{\widehat{M}} + \frac{\alpha}{n}\left\|\sum_{i=1}^n X_i\varepsilon_i\right\|_{\widehat{M}}$$

$$\leq (1 - \alpha\widehat{\lambda}_{\min}^+)\|w_t^\star - \tilde{w}_t^\star\|_{\widehat{M}} + \frac{\alpha}{n}\left\|\sum_{i=1}^n X_i\varepsilon_i\right\|_{\widehat{M}}. \tag{4}$$

where in the step $(a)$ we note that $\widehat{M}(I - \alpha\widehat{\Sigma})(w_t^\star - \tilde{w}_t^\star) = \widehat{M}(I - \alpha\widehat{\Sigma})\widehat{M}(w_t^\star - \tilde{w}_t^\star)$ (since $\widehat{M}^2 = \widehat{M}$ and $\widehat{\Sigma}\widehat{M} = \widehat{\Sigma}$).

Now we use the fact that an elementary recursive relation $x_{t+1} \leq a_t x_t + b_t$ with $x_0 = 0$ unwinds to $x_T \leq \sum_{t=1}^T b_t \prod_{k=t+1}^T a_k$, which gives

$$\|w_T^\star - \tilde{w}_T^\star\|_{\widehat{M}} \leq \frac{\alpha}{n}\left\|\sum_{i=1}^n X_i\varepsilon_i\right\|_{\widehat{M}} \sum_{t=1}^T (1 - \alpha\widehat{\lambda}_{\min}^+)^{T-t}$$

$$\leq \frac{\alpha}{n}\left\|\sum_{i=1}^n X_i\varepsilon_i\right\|_{\widehat{M}} \frac{1 - (1 - \alpha\widehat{\lambda}_{\min}^+)^T}{\alpha\widehat{\lambda}_{\min}^+}.$$

Thus,

$$\frac{2}{n}\sum_{i=1}^n \mathbb{E}_{\boldsymbol{\varepsilon}}\left[\varepsilon_i\left(w_T^\star - w_0^\star\right)^\top X_i\right] \leq \frac{2}{n}\cdot\frac{1}{n}\mathbb{E}_{\boldsymbol{\varepsilon}}\left[\left\|\sum_{i=1}^n X_i\varepsilon_i\right\|^2 \frac{1}{\widehat{\lambda}_{\min}^+}\right]$$

$$\leq \frac{2\sigma^2}{n}\cdot\frac{1}{\widehat{\lambda}_{\min}^+}$$

where we used a basic fact that

$$\mathbb{E}_{\boldsymbol{\varepsilon}}\left[\left\|\sum_{i=1}^n X_i\varepsilon_i\right\|^2 \,\Big|\, X_1,\ldots,X_n\right] = \sigma^2\sum_{i=1}^n \|X_i\|^2 \leq \sigma^2 n.$$

Putting all together completes the proof. □

## D  Concentration of the Smallest Non-zero Eigenvalue: Proof

Here we show a non-asymptotic concentration of the smallest positive eigenvalue as stated in Section 3.3.

**Lemma 1 (restated).** *Let $\boldsymbol{D}^\top = [\boldsymbol{X}_1, \ldots, \boldsymbol{X}_n] \in \mathbb{R}^{d \times n}$ be a matrix with i.i.d. columns, such that $\max_i \|\boldsymbol{X}_i\|_{\psi_2} \leq K$, and let $\widehat{\boldsymbol{\Sigma}} = \boldsymbol{D}^\top \boldsymbol{D}/n$, and $\boldsymbol{\Sigma} = \mathbb{E}[\boldsymbol{X}_1 \boldsymbol{X}_1^\top]$. Then, for every $x \geq 0$, with probability at least $1 - 2e^{-x}$, we have*

$$\lambda_{\min}^+(\widehat{\boldsymbol{\Sigma}}) \geq \lambda_{\min}^+(\boldsymbol{\Sigma}) \left( 1 - K^2 \left( c\sqrt{\frac{d}{n}} + \sqrt{\frac{x}{n}} \right) \right)_+^2 \qquad \text{for } n \geq d \,,$$

*and furthermore, assuming that $\|\boldsymbol{X}_i\|_{\boldsymbol{\Sigma}^\dagger} = \sqrt{d}$ a.s. for all $i \in [n]$, we have*

$$\lambda_{\min}^+(\widehat{\boldsymbol{\Sigma}}) \geq \lambda_{\min}^+(\boldsymbol{\Sigma}) \left( \sqrt{\frac{d}{n}} - K^2 \left( c + 6\sqrt{\frac{x}{n}} \right) \right)_+^2 \qquad \text{for } n < d \,,$$

*where we have an absolute constant $c = 2^{3.5}\sqrt{\ln(9)}$.*

Lemma 1 proven shortly, is based upon the the next theorem gives us a non-asymptotic version of Bai-Yin law [Bai and Yin, 1993] for rectangular matrices whose rows are sub-Gaussian isotropic random vectors.

**Theorem 4** ([Vershynin, 2012, Theorem 5.39]). *Let $\boldsymbol{A} \in \mathbb{R}^{n \times d}$ whose rows $(\boldsymbol{A}^\top)_i$ are independent sub-Gaussian isotropic random vectors in $\mathbb{R}^d$, such that $K = \max_{i \in [n]} \|(\boldsymbol{A}^\top)_i\|_{\psi_2}$. Then for every $x \geq 0$, with probability at least $1 - 2e^{-x}$ one has*

$$\sqrt{n} - 2^{3.5} K^2(\sqrt{\ln(9)d} + \sqrt{x}) \leq s_{\min}(\boldsymbol{A}) \leq s_{\max}(\boldsymbol{A}) \leq \sqrt{n} + 2^{3.5} K^2 \left( \sqrt{\ln(9)d} + \sqrt{x} \right) \,.$$

**Theorem 5** ([Vershynin, 2012, Theorem 5.58]). *Let $\boldsymbol{A} \in \mathbb{R}^{d \times n}$ whose columns $\boldsymbol{A}_i$ are independent sub-Gaussian isotropic random vectors in $\mathbb{R}^d$ with $\|\boldsymbol{A}_i\| = \sqrt{d}$ a.s., such that $K = \max_{i \in [n]} \|\boldsymbol{A}_i\|_{\psi_2}$. Then for every $x \geq 0$, with probability at least $1 - 2e^{-x}$ one has*

$$\sqrt{d} - 2^{3.5} K^2(\sqrt{\ln(9)n} + 6\sqrt{x}) \leq s_{\min}(\boldsymbol{A}) \leq s_{\max}(\boldsymbol{A}) \leq \sqrt{d} + 2^{3.5} K^2(\sqrt{\ln(9)n} + 6\sqrt{x})$$

Above two theorems lead to the following non-asymptotic version of a Bai-Yin law.

*Proof of Lemma 1.* The proof considers two cases: 1) when number of observations exceeds the dimension, which is handled by the concentration of a minimal non-zero eigenvalue of a covariance matrix; 2) when dimension exceeds number of observations, which is handled by concentration of the Gram matrix.

**Case $n \geq d$.** We will apply Theorem 4 with $\boldsymbol{A} = (\boldsymbol{\Sigma}^{\dagger \frac{1}{2}} \boldsymbol{D}^\top)^\top = \boldsymbol{D} \boldsymbol{\Sigma}^{\dagger \frac{1}{2}}$ whose rows are independent and isotropic, and in addition by Cauchy-Schwarz inequality:

$$\|\boldsymbol{\Sigma}^{\dagger \frac{1}{2}}\| s_{\min}(\boldsymbol{D}) \geq s_{\min}(\boldsymbol{A}) \geq \sqrt{n} - 2^{3.5} K^2(\sqrt{\ln(9)d} + \sqrt{x})$$

with probability at least $1 - e^{-x}$ for $x > 0$. Observing that $\|\boldsymbol{\Sigma}^{\dagger \frac{1}{2}}\| = s_{\min}^+(\boldsymbol{\Sigma})^{-1/2}$, this implies that

$$s_{\min}(\boldsymbol{D}) \geq \sqrt{s_{\min}^+(\boldsymbol{\Sigma})} \left( \sqrt{n} - 2^{3.5} K^2 \left( \sqrt{\ln(9)\,d} + \sqrt{x} \right) \right) \,,$$

while dividing through by $\sqrt{n}$, taking the non-negative part of the r.h.s. and squaring gives us

$$\lambda_{\min}(\widehat{\boldsymbol{\Sigma}}) \geq \lambda_{\min}^+(\boldsymbol{\Sigma}) \left( 1 - 2^{3.5} K^2 \left( \sqrt{\ln(9)\frac{d}{n}} + \sqrt{\frac{x}{n}} \right) \right)_+^2 \,.$$

**Case $n < d$.** In this case we essentially study concentration of a smallest singular value of a Gram matrix $\widehat{\boldsymbol{G}} = \frac{1}{d} \boldsymbol{D} \boldsymbol{D}^\top$. For the case $n < d$, Theorem 4 would give us a vacuous estimate, and therefore we rely on Theorem 5 which requires additional assumption that columns of $\boldsymbol{D}^\top$ lie on a (elliptic)

sphere of radius $\sqrt{d}$. In particular, similarly as before, applying Theorem 5 to the matrix $\boldsymbol{\Sigma}^{\dagger\frac{1}{2}}\boldsymbol{D}^{\top}$ with isotropic columns $\boldsymbol{\Sigma}^{\dagger\frac{1}{2}}\boldsymbol{X}_i$ satisfying $\|\boldsymbol{\Sigma}^{\dagger\frac{1}{2}}\boldsymbol{X}_i\| = \sqrt{d}$ a.s. for all $i \in [n]$, we get

$$\|\boldsymbol{\Sigma}^{\dagger\frac{1}{2}}\| s_{\min}(\boldsymbol{D}^{\top}) \geq s_{\min}\left(\boldsymbol{\Sigma}^{\dagger\frac{1}{2}}\boldsymbol{D}^{\top}\right) \geq \sqrt{d} - 2^{3.5}K^2(\sqrt{\ln(9)n} + 6\sqrt{x})$$

with probability at least $1 - e^{-x}$ for $x > 0$. Again, this gives us

$$s_{\min}(\boldsymbol{D}) \geq \sqrt{s_{\min}^+(\boldsymbol{\Sigma})}\left(\sqrt{d} - 2^{3.5}K^2\left(\sqrt{\ln(9)\,n} + 6\sqrt{x}\right)\right) ,$$

while dividing through by $\sqrt{d}$, taking the non-negative part of the r.h.s. and squaring gives us

$$\lambda_{\min}(\widehat{\boldsymbol{G}}) \geq \lambda_{\min}^+(\boldsymbol{\Sigma})\left(1 - 2^{3.5}K^2\left(\sqrt{\ln(9)\frac{n}{d}} + 6\sqrt{\frac{x}{d}}\right)\right)_+^2 .$$

Now we relate $\lambda_{\min}(\widehat{\boldsymbol{G}})$ to the smallest non-zero eigenvalue of $\widehat{\boldsymbol{\Sigma}}$ (see also [Bai and Yin, 1993, Remark 1]). The smallest eigenvalue of $d\widehat{\boldsymbol{G}}$ corresponds to $d - n + 1$-th smallest eigenvalue of $n\widehat{\boldsymbol{\Sigma}}$, that is $d\lambda_{\min}(\widehat{\boldsymbol{G}}) = n\lambda_{\min}^+(\widehat{\boldsymbol{\Sigma}})$. That said, multiplying the previous inequality through by $d/n$ and rearranging, we get

$$\lambda_{\min}^+(\widehat{\boldsymbol{\Sigma}}) \geq \lambda_{\min}^+(\boldsymbol{\Sigma})\left(\sqrt{\frac{d}{n}} - 2^{3.5}K^2\left(\sqrt{\ln(9)} + 6\sqrt{\frac{x}{n}}\right)\right)_+^2$$

The proof is now complete. $\qquad\square$

## E  Minimum eigenvalue and condition number

Previous works considered the link between the condition number of the features and the DD behavior [Rangamani et al., 2020]. In this work, the analysis focuses more particularly on the minimum eigenvalue. In the following small experiments, we empirically show that in the experiments shown in the main paper, the condition number is driven by the minimum eigenvalue, and that the maximum eigenvalue stays close to a constant order when we increase the size of the features. In Figure 4, we use the same setting as in the MNIST experiment in Figure 2 is the main paper. We obseve that the behavior of the condition number follows the minimum eigenvalue, while the maximum eigenvalue stays between 10 and 100 as we increase the width of the networks.

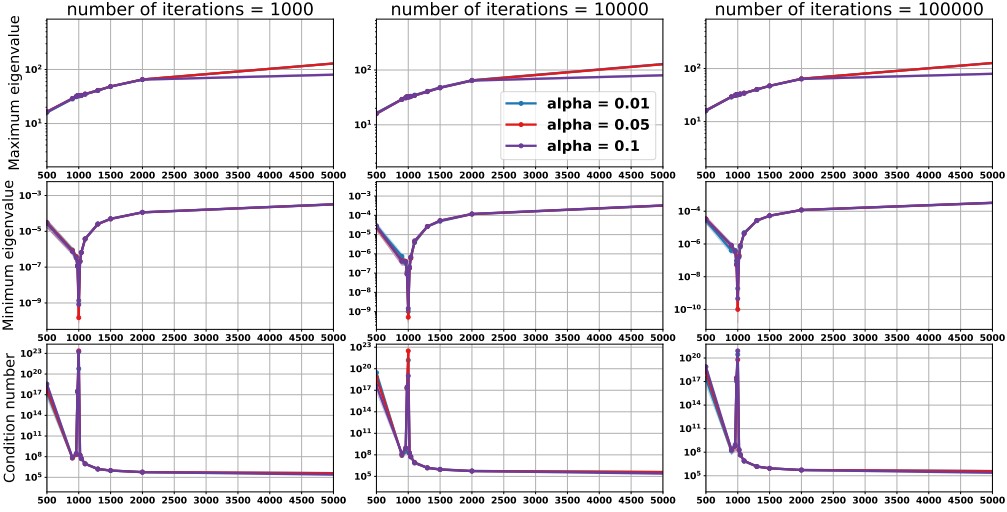

Figure 4: Maximum and minimum eigenvalues and condition numbers of the features of one hidden layer networks of variable width: MNIST - 1000 samples for training, networks trained with gradient descent and different step sizes.

# F   More on the effect of depth

In section 6, we suggested that the ill-conditioning of the intermediary features of a neural network is not only due to the size of the network, but also to the weights distribution across the layers. More particularly, we suggest here that the optimization difficulty we observe for deep neural networks is linked among other factors to the minimum eigenvalue of the activations of the penultimate layer.

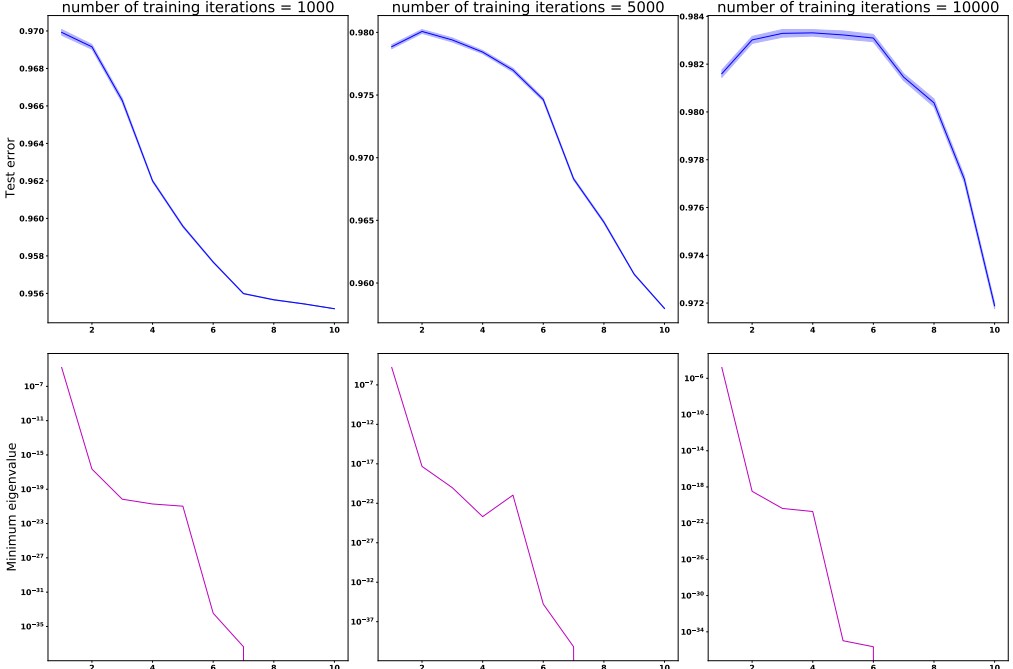

Figure 5: Mean test error and minimum eigenvalue for networks of fixed width = 500 and varying depth: MNIST - 1000 samples for training, 10000 samples for test, networks trained with gradient descent and step size 0.01.

To support our hypothesis, we run an experiment where we train networks of a fixed width (equal to 500) and depth varying from 2 to 10. We track the test error at various stages of training and the minimum eigenvalue of the features of the last layer. In Figure 5, we can observe that as expected, the deeper the network, the harder it is to train them. This is reflected in the increasing test error. For the deepest network, simple gradient descent fails to obtain a reasonable performance even after 10000 iterations. Moreover, we observe that the deeper the network, the smaller is the minimum eigenvalue, and the most ill-conditioned settings get even worse with training.

To further this analysis, we also compare networks with 3 hidden layers where we increase the width in all the layers and in the penultimate layer only, creating bottleneck in the earlier layers. This experiment complements Figure 3. In Figure 6, we observe that the bottleneck results in a more important drop in the eigenvalue around the width 1000 (width of the last layer in this case). Moreover, the minimum eigenvalue stays smaller than the other considered architectures when we increase the depth. This is reflected in a higher test error, confirming once more the effect of the conditioning of the last layer features on the final performance of the network when trained with gradient descent.

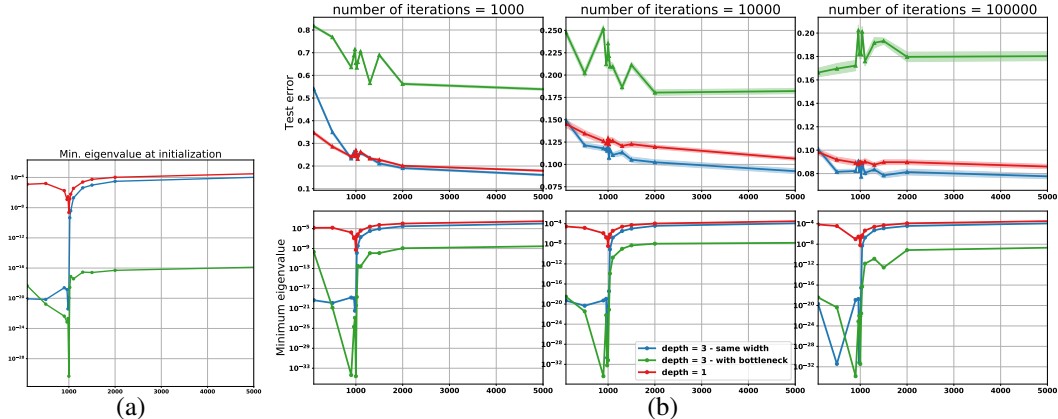

Figure 6: Training networks of increasing width with 1 and 3 hidden layers on MNIST - For the version with bottleneck, only the size of the last hidden layer is increased, while the other layers are composed of 10 neurons: (a) Minimum positive eigenvalue of the intermediary features at initialization - (b) Test error and corresponding minimum eigenvalue of the intermediary features at different iterations

# G    Additional Empirical Evaluation

## G.1    Experimental settings - More details

In our experiments, we considered two datasets: MNIST and FashionMNIST. Both datasets have an input dimension of $784$ and a training set of $6.10^4$ samples. As our theory predicts that the drop in the minimum eigenvalue and the performance of the models happens when the feature size reaches the size of the training set, and in order to keep our model tractable, we use subsets of size 1000 of the training sets. These subsets are randomly chosen and kept the same when the size of the model increases. All the models are trained with plain gradient descent, with a fixed step size. We use a step size of $0.01$ unless stated otherwise. All the weights of the networks are initialized from a truncated normal distribution with a scaled variance. Finally, for the MNIST experiment in Figure 2, the mean and standard errors are estimated from runs with different seeds. For the other experiments, the mean and standard errors of the test error are estimated by splitting the test set into 10 subsets.

## G.2    More on the effect of architectural choices

In section 6, we suggested that for neural networks the quantity of interest might also be $\widehat{\lambda}_{\min}^+$ for intermediary features, which is affected by size of the model but also by the distribution of the weights and architectural choices. Section F shows some experiments that validate this hypothesis. To further our analysis, we question here the impact of skip connections on the eigenvalue of features at initialization. The difficulty that depth cause for the optimization of neural networks led to our reliance on skip connections among other tricks [De and Smith, 2020]. Here, we hypothesize that skip connections make the optimization of deep networks easier thanks to a better conditioning of the feature, through a less severe drop in the minimum eigenvalue around the interpolation threshold. Figure 7 shows that for a deep network with skip connection, the minimum eigenvalue of the penultimate layer activations behaves like this of a shallow neural network.

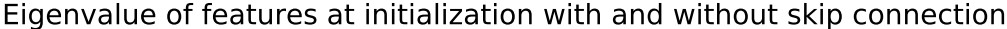

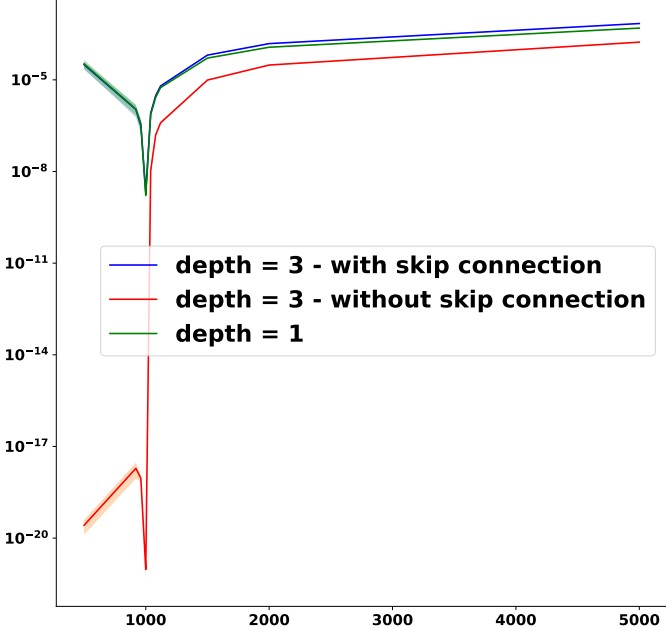

Figure 7: Mean minimum eigenvalue at initialization for networks of depths 1 and 3 and varying width. For the network of depth 3, we show two variants: with and without skip connection. The skip connection makes the deep network eigenvalue behave like the shallow network's.

### G.3 On the position of the peak

The jump of the test error predicted by our theory and observed in our experiments appears when the width of the penultimate layer of the network reaches the size of the training set. The jump of the test loss observed in Belkin et al. [2019] appears when the size of the model reaches the size of the training set multiplied by the number of classes. It is then a natural question to ask whether these two positions coincide. Let us consider a network with a single hidden layer for simplicity. For an input of dimension $d$, a hidden layer of width $h$ and a problem with $k$ classes, the size of such a network is $(d + 1)h + (h + 1)k$. For a training set of size $n$, Belkin et al. [2019] observes a peak in the test loss when $(d + 1)h + (h + 1)k = nk$, which corresponds to a model with a hidden layer of width $h = \frac{nk-k}{d+k+1}$. When training with 1000 samples from MNIST (as in our experiments), this width is then h $\approx$ 12. Recall that the peak highlighted here appears at $h = 1000$, which is a much bigger model. This observation reinforces the observation that neural networks show not only a double but a triple [d'Ascoli et al., 2020] (or a multiple?) descent behavior. A thorough verification is however beyond the scope of this work, and is left to future research.