# OpenReview forum: "On the Role of Optimization in Double Descent: A Least Squares Study"
_NeurIPS.cc/2021/Conference — NeurIPS 2021 Poster_

### Official Review · Reviewer_uq7X · 2021-07-12

**Rating:** 8
**Confidence:** 4

**Summary:**

This paper deals with the double descent phenomenon in the setting of least-squares regression by gradient descent. Similarly to a number of recent works, it shows a double descent behavior for the excess risk: a U shaped curve until the number of features coincides with the number of datapoints. The main contribution of the paper is to relate this peak around interpolation to the learning dynamics of the problem at hand. More precisely, the main result (Theorem 1) is an upper bound on the excess risk where the dynamics of learning are explicitly present. This result is specialized to the case of least-squares with random design in the noiseless setting (Theorem 2) and with Gaussian noise independent from the inputs (Theorem 3). In both these cases, the dominating term is controlled by the smallest non-zero singular value of the sample covariance matrix. Therefore, under some assumption on the data, one can get precise asymptotics by using the Bai-Yin result. This is done in Section 2.3 in a non-asymptotic fashion. The paper concludes by an empirical study on neural networks with one and 3 hidden layers, showing a link between the smallest eigenvalue of the covariance of the intermediary features at initialization and the test error.

**Limitations And Societal Impact:**

yes

**Main Review:**

***Main comments***

The paper is overall very interesting, and well-written. The proofs are correct as far as I can tell, I have only one small technical question detailed below. I think that the link between optimization and double descent was not addressed by previous work and is a very exciting new avenue of research. The precise expressions obtained in the paper depend on strong assumptions and in the particular case of least-squares. These are the main limitations of the paper. The empirical argument does not seem so strong.  More detailed comments follow:

 - statement of Theorem 2: the reason for a high probability statement in Theorem 2 is unclear to me. Maybe I am missing something here, but the only sources of randomness here are S (the sampling) and W_0 (the initialization). As far as I understand the proof of Theorem 1, from which Theorem 2 is a consequence, the expectation is taken with respect to both these sources of randomness. Therefore it seems that this statement should not be there. This statement is also not present in the Appendix. The same question holds for Theorem 3. I will increase my score if this is properly addressed by the authors.

 - l. 47: my understanding of Belkin et al. '19 is that the test error *can* be lower in the overparametrized regime, but not necessarily.

 - stochastic gradient descent: the main optimization tool for recent architectures is SGD. Can the present analysis be extended to SGD instead of GD? The main obstruction seems to be proving that SGD is (\Delta,\Mhat)-admissible, with of course another \Delta. What would be \Delta in that case.

***Typos and minor comments***

 - l. 211: parammetrization -> parametrization
 - Figure 2 and 3 are hard to read, I would suggest using bigger fonts for the x- and y-axis.

***After rebuttal***

The authors have properly addressed my concerns, I am therefore increasing my score.

**Time Spent Reviewing:**

10

---

> ### Author Response · Authors · 2021-08-10
> **Response to Reviewer uq7X**
>
> * Comment _"statement of Theorem 2: the reason for a high probability statement in Theorem 2 is unclear to me"_
>   * Thank you for pointing this out, this is a typo. In both Theorem 2 and 3 we condition on covariates $X_1, \ldots, X_n$, that is expectation is a conditional expectation, and so "w.h.p. over $S$" is removed from the statement.
>
> * Comment _"Can the present analysis be extended to SGD instead of GD?"_
>   * Yes, in this case the algorithm is randomized, not only in $W_0$ (as it is now), but also in the indices of chosen examples: let's call these indices $\xi$. In such case we would verify Definition 1 for the algorithm ($f$) with $\xi$ fixed, and eventually take an expectation (we already do so for $W_0$). One straightforward idea would be to use the fact that, given a fixed $\xi$, and appropriate step size schedule, SGD updates for convex (or strongly convex) losses are non-expansive, see [1] Sec 3.2. We suspect that such analysis would give $\Delta$ polynomially decreasing in $T \hat{\lambda}_{\min}^+$ (rather than exponentially).
>
> * Comment on line 47: Thanks for the remark! Indeed, we will discuss this point.
>
> $[1]$ Hardt, Moritz, Ben Recht, and Yoram Singer. "Train faster, generalize better: Stability of stochastic gradient descent." ICML 2016.
>
> Thank you for your feedback and for the time spent on our work!

---

### Official Review · Reviewer_bWET · 2021-07-14

**Rating:** 7
**Confidence:** 3

**Summary:**

This paper studies the double descent behaviour in the context of least squares, using a theoretical analysis of the spectrum of the feature covariance matrix. It conjectures that the double descent (in this case) is a consequence of the behaviour of the smallest positive eigenvalue.

**Limitations And Societal Impact:**

Yes

**Main Review:**

In summary the paper is very well written and flows nicely. The double descent behaviour remains a major open question un deep learning theory. Although the analysis in this paper is restricted to the least squares case, it is still a valuable contribution to the litterature.

Details and questions:

- The main result is that the U-shaped curve of the smallest positive eigenvalue of the feature covariance matrix is the reason behind the double descent behaviour (in the least squares case), this is reflected in the generalization bound provided in the paper. The empirical results confirm the theoretical claims.

- Extending the analysis to deep networks might not be feasable, but I believe that using a simplification of deep nets such as the NTK regime (linearized) might be a very nice addition to this paper.

- Recent works have shown a triple descent behaviour for deep nets (e.g. d'Ascoli et al. Triple descent and the two kinds of overfitting: Where & why do they appear?). Could the authors comment on how the smallest positiove eigenvalue behaves in this case? (since the U shaped curve would no longer explain the triple descent )

**Time Spent Reviewing:**

2 hours

---

> ### Author Response · Authors · 2021-08-10
> **Response to Reviewer bWET**
>
> * Applicability to neural networks:
>   * As reviewer bWET point out, one possibility for adapting our results to the neural network learning, would be to consider a Neural Tangent Kernel (NTK) framework. Here we would need to rely on the concentration of the random feature sample covariance matrix, and for the shallow networks, concentration phenomena are expected to be similar as presented in our work. For the deep networks, the spectrum of the NTK matrix is not well understood and this is a cutting-edge area of research itself. Obviously, the jury is still out on whether the NTK is a good theory for the neural network learning.
>   * Alternatively to NTK, one could think of this problem as a "feature-map" plus "linear predictor" combination. To an extent, this is what we pursued in our experiments involving DNNs. In such a case, we need to understand the concentration of the spectrum of a sample covariance matrix formed from the features: This is akin to studying the spectrum of NTK of a deep network, and as mentioned before, this is currently an active area of research. We hope to be able to investigate this connection further as future work, but believe the findings presented in the current work are can be meaningful for the community.
>
> * Interpretation of the triple-descent:
>   * Indeed, triple descent was identified in a number of works, such as you mention (we will definitely cite it) and (Adlam and Pennington, 2020). We believe that considering shallow neural nets where only the hidden layer is trained (output layer is fixed), the smallest positive eigenvalue should behave similarly as we discuss in this submission (concentration of spectrum of the random feature sample covariance matrix). However, when training all parameters jointly in deeper networks, the behaviour of the eigenvalue might be very different depending on the configuration of the network. We suspect that increasing the width in certain layers while keeping others fixed, might introduce additional peaks. Loosely speaking this might happen because adjusting number of parameters in certain layers changes the conditioning of a feature covariance matrix (we briefly discuss the possibility of studying DD in neural nets throughout the "feature-map" point of view discussed earlier).
>
> Thank you for your feedback and for the time spent on our work!

---

### Official Review · Reviewer_bMDY · 2021-07-15

**Rating:** 7
**Confidence:** 2

**Summary:**

This paper proves an excess risk bound for linear least squares regression problems in an overparametrized setting which takes into account (a proxy of) the condition number of the optimization when done via Gradient Descent. The bound allows to shed a new light on the Double Descent phenomenon.

**Limitations And Societal Impact:**

This work is theoretical and there are no such considerations here.

**Main Review:**

The main originality here is to argue that DD is a result of optimizing the parameters using gradient descent and that its precise form depends mainly on the minimal positive eigenvalue of the sample covariance matrix.

This point of view is, in my knowledge, new and the bounds shown are sharp enough to allow some meaningful analysis. In particular, there seems to be a nice fit between predictions of the theory and the empirical results.

Regarding presentation, I think the writing could be improved. The text is quite dense and the paper wasn't easy to read in some places. I also think that theorems 2 and 3 could be summarized as the same result. This would free some space to transfer some experiments and discussions from the appendix.

My main problem with this work, which is also acknowledged by the authors at the last section, is its lack of applicability to deep neural architectures: while the tools of analysis are presented as being potentially useful for modern models (in the abstract and in the introduction), nothing here points to the truth of that and I don't see how the same type of bounds could be proved for actual DNNs.

Another more minor point is the dependence on the number of steps T: It is known that there is a DD phenomenon wrt T, and this is not represented in the proven bounds as I understand them.

Edit after rebuttal: I thank the authors for their answer to my review. I hope they will be able to incorporate them in the camera ready version of the paper. I think this is a good contribution and raise my score accordingly.

**Time Spent Reviewing:**

2

---

> ### Author Response · Authors · 2021-08-10
> **Response to Reviewer bMDY**
>
> * Applicability to neural networks:
>   * As reviewer bWET point out, one possibility for adapting our results to the neural network learning, would be to consider a Neural Tangent Kernel (NTK) framework. Here we would need to rely on the concentration of the random feature sample covariance matrix, and for the shallow networks, concentration phenomena are expected to be similar as presented in our work. For the deep networks, the spectrum of the NTK matrix is not well understood and this is a cutting-edge area of research itself. Obviously, the jury is still out on whether the NTK is a good theory for the neural network learning.
>   * Alternatively to NTK, one could think of this problem as a "feature-map" plus "linear predictor" combination. To an extent, this is what we pursued in our experiments involving DNNs. In such a case, we need to understand the concentration of the spectrum of a sample covariance matrix formed from the features: This is akin to studying the spectrum of NTK of a deep network, and as mentioned before, this is currently an active area of research. We hope to be able to investigate this connection further as future work, but believe the findings presented in the current work are can be meaningful for the community.
>
> * Comment: _"It is known that there is a DD phenomenon wrt T, and this is not represented in the proven bounds as I understand them."_
>   * Previous literature considers analytic solution to the least-squares problem and therefore $T$ in such theories is not present (one can think of the Moore-Penrose pseudoinverse as GD with $T \to \infty$). In this work we investigate a regime where $T$ is finite and thus show how DD depends on $T$: In particular, DD caused by optimization sharpens-up as $T$ increases.
>
> * Feedback regarding improving the presentation:
>   * We will try to integrate these suggestions.
>
> Thank you for your feedback and for the time spent on our work!

---

### Author Response · Authors · 2021-08-10
**Comment to All Reviewers**

We thank the reviewers for the time spent on our submission and for their feedback.

We first provide a general response addressing some common feedback, and then we address each reviewers' feedback separately.

Overall we are encouraged by the reviewers' reactions to our work and their scores. Reviewers bWET and uq7X wrote that our paper "is very well written and flows nicely" and "is overall very interesting, and well-written" ---we thank them for these positive comments. On the other hand, we take note that reviewer bMDY wrote feedback about the presentation being dense and not easy to follow ---we are open to make an effort to improve the presentation, but pointers to specific parts of the paper would help us to address this concern.

Several reviewers felt that a weakness of our work is that our theoretical results do not carry over to neural network learning. As reviewer bWET pointed out, one possibility for adapting our results to the setting of neural network learning, would be to consider the Neural Tangent Kernel (NTK) framework. Here we would need to rely on the concentration of the random feature sample covariance matrix, and for the shallow networks, concentration phenomena are expected to be similar as that presented in our work. For the deep networks, the spectrum of the NTK matrix is not well understood and this is a cutting-edge area of research itself. Obviously, the jury is still out on whether the NTK is a good theory for the neural network learning.

Alternatively to NTK, one could think of this problem as a "feature-map" plus "linear predictor" combination. To an extent, this is what we pursued in our experiments involving DNNs. In such a case, we need to understand the concentration of the spectrum of a sample covariance matrix formed from the features: This is akin to studying the spectrum of NTK of a deep network, and as mentioned before, this is currently an active area of research.

We think our work contributes an interesting perspective on the DD phenomena, by connecting the peak of DD to the optimization. It seems that all reviewers have reacted positively to this connection. We hope to be able to investigate this connection further as future work. While we agree that our theoretical contribution is limited to the specific setting of linear least squares learning, we think that the combination of the insights from this setting and the results of our experiments with neural network learning present an original perspective on DD on which further work can be built, and we believe the findings presented in the current work can be meaningful for the community.

---

### Decision · Program_Chairs · 2021-09-28

**Decision:**

Accept (Poster)

**Comment:**

This paper considers the double descent phenomenon for least-squares regression with gradient descent optimization, showing a double descent behavior for the excess risk, and relating this peak around interpolation to the learning dynamics of the optimization algorithm.  All reviewers agreed that the paper made a useful contribution for the community.

**Consistency Experiment:**

NeurIPS has a long history of experimentation. In 2014, NeurIPS ran an experiment in which 10% of submissions were reviewed by two independent committees to quantify the randomness in the review process. This year, we repeated a variant of this experiment to see how the quality of the review process has changed over time.  This paper was part of the experiment and was therefore assigned to two committees (consisting of reviewers, an Area Chair, and a Senior Area Chair) that reached independent decisions.  If both committees made the same recommendation, this recommendation was followed. If a single committee recommended acceptance, the paper was accepted (with the exception of a few cases in which the other committee identified what we considered a fatal flaw, e.g., an error in a key result).

This copy’s committee reached the following decision: **Accept (Poster)**

The other committee assigned to the paper recommended **Reject**.  You can find the other set of reviews, along with any follow up discussion with the authors here:
https://openreview.net/forum?id=zzSVN5x8JiX1m